# Toward effective protection against diffusion based mimicry through score distillation

**Haotian Xue** [1]    **Chumeng Liang** [*2,3]    **Xiaoyu Wu** [*2]    **Yongxin Chen** [1]

[1] Georgia Institute of Technology
[2] Shanghai Jiao Tong University
[3] University of Southern California

## ABSTRACT

While generative diffusion models excel in producing high-quality images, they can also be misused to mimic authorized images, posing a significant threat to AI systems. Efforts have been made to add calibrated perturbations to protect images from diffusion-based mimicry pipelines. However, most of the existing methods are too ineffective and even impractical to be used by individual users due to their high computation and memory requirements. In this work, we present novel findings on attacking latent diffusion models (LDM) and propose new plug-and-play strategies for more effective protection. In particular, we explore the bottleneck in attacking an LDM, discovering that the encoder module rather than the denoiser module is the vulnerable point. Based on this insight, we present our strategy using Score Distillation Sampling (SDS) to double the speed of protection and reduce memory occupation by half without compromising its strength. Additionally, we provide a robust protection strategy by counterintuitively minimizing the semantic loss, which can assist in generating more natural perturbations. Finally, we conduct extensive experiments to substantiate our findings and comprehensively evaluate our newly proposed strategies. We hope our insights and protective measures can contribute to better defense against malicious diffusion-based mimicry, advancing the development of secure AI systems. Codes for this paper are available in https://github.com/xavihart/Diff-Protect.

## 1 INTRODUCTION

Generative Diffusion Models (GDMs)  (Song et al., 2020b; Ho et al., 2020) have achieved remarkable success in the realm of image synthesis and editing tasks. One lurking concern is that abusers may utilize well-trained GDMs to generate digital mimicry of other individuals: doing GDM-based inpainting maliciously on photos of the victim  (Zhang et al., 2023a), or appropriating the styles of an artist without any legal consent (Andersen, 2023; Setty, 2023). In the absence of protections over images, GDMs may be easily turned toward less ethical applications.

Current efforts have been directed toward safeguarding unauthorized images from diffusion-based mimicry in the context of adversarial attacks. By introducing perturbations within a limited budget, they can deceive diffusion models to produce chaotic results. AdvDM  (Liang et al., 2023) try to generate adversarial examples for the diffusion model in a general way by attacking the noise prediction module. Photoguard (Salman et al., 2023) and Glaze  (Shan et al., 2023) focus on minimizing the distance in the latent space between the projected image and a prepared target style. Mist (Liang & Wu, 2023) combines semantic loss and textural loss with a discussion on the choice of target textural pattern, showing promising results in protection against mimicry. While all these methods can achieve good performance in certain tasks (e.g. image-to-image, image-to-style), some key problems remain to be solved: (1) **Heavy Computational Cost**: when attacking the GDM, we need to calculate the gradient of output images over the input of the GDM  (Liang et al., 2023; Liang & Wu, 2023; Salman et al., 2023), whose computational demand can impose a burden, particularly on individual users. (2) **Insufficiently Explored Design Space**: In their work,  (Liang & Wu, 2023)

---

Correspondence to: htxue.ai@gatech.edu, the second and third author contribute equally

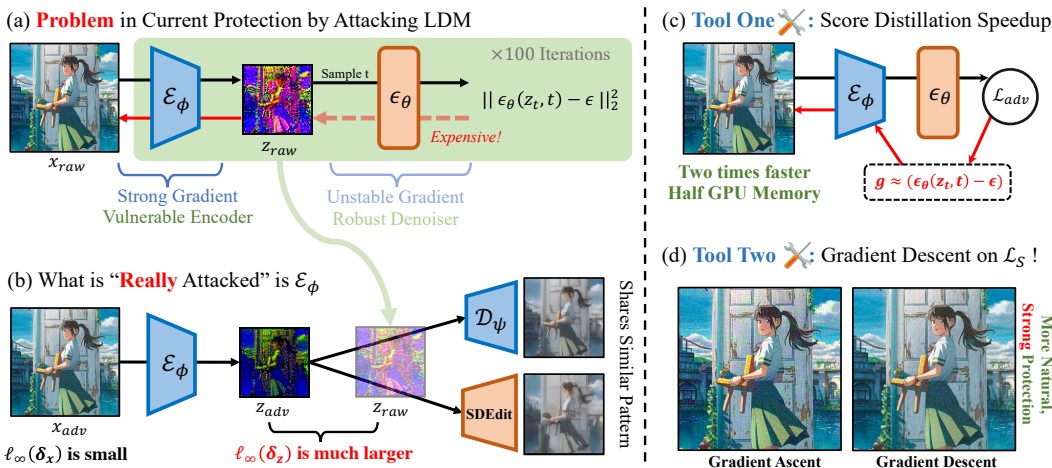

Figure 1: **What Should We Focus On When Protecting Against Diffusion-based Mimicry?** (a) Generating adversarial samples for LDMs is expensive with over 100 steps of backpropagation over denoiser $\epsilon_\theta$. The gradient of the denoiser tends to be really weak and unstable, compared with the strong gradient attacking the encoder, showing that $\epsilon_\theta$ is much more robust than the encoder $\mathcal{E}_\phi$. (b) After the PGD-iterations, the latent $z$-space has a much larger perturbation than the $x$-space, indicating $\mathcal{E}_\phi$ accounts for the effectiveness of the attack. (c, d) Our proposed design space, with much better efficiency and flexibility against three kinds of mimicry.

outlined the design space, encompassing textural loss (on the encoder side) and semantic loss (on the denoising module side). However, the rationale and mechanisms behind the effectiveness of each component remain unexplained.

In this work, we investigate the **bottleneck** of the attack against diffusion models, and demonstrate that the primary impact of attacks on the latent diffusion model (LDM) actually results from the vulnerable encoder. Based on these key discoveries, we propose the design space of more effective protection against diffusion-based mimicry, focusing on the LDM (Rombach et al., 2022). By introducing Score Distillation Sampling (SDS) (Poole et al., 2022) into the adversarial optimization process, we dramatically reduce the high requirement for computational resources without sacrificing effectiveness. Following Mist (Liang & Wu, 2023), we provide a more in-depth exploration of the optimization's design space. In particular, we highlight a series of intriguing properties of attacking an LDM: (1) Latent space is the bottleneck: while the semantic loss focuses on attacking the overall LDM, we found that the fulcrum is actually on the image encoder. We have designed comprehensive experiments to show that the encoder is more vulnerable while the denoising module is robust (Figure 1 (a)). (2) Both maximizing and minimizing the semantic loss can bring reasonable attacks, and the latter can bring more imperceptible attacks. While previous work Liang et al. (2023) shows that maximizing the semantic loss can fool the LDM, we show that minimizing the semantic loss can also fool the LDM by blurring the output, with a more natural perturbation attached. Actually, the perturbation's naturalness holds significance to ensure that the overall user experience remains uncompromised.

We conduct extensive experiments to support our arguments above. Following (Salman et al., 2023; Liang et al., 2023), we conduct experiments on i) global image-to-image edit, ii) image-to-image inpainting, and iii) textual inversion. While (Liang et al., 2023) focuses more on artworks and (Salman et al., 2023) focuses more on portraits and realistic photographs, our evaluations encompass a broader spectrum of contents that may suffer from potential malicious mimicry: including anime, portraits, artworks, and landscape photos. Our main contributions are listed below:

1. We reveal the bottleneck of attacks against LDMs, showing that the encoder is **much more vulnerable** than the denoiser. (Section 4)

2. We propose a more effective protection framework to generate perturbations against diffusion-based mimicry by introducing Score Distillation Sampling into the optimization, dramatically reducing the computational burden by 50%. (Section 5.1)

3. We are the first to systematically explore the design space of the attacks against LDM. We found two possible directions of attacks by maximizing and minimizing the semantic loss. The latter results in more imperceptible perturbation with competitive protection effects. (Section 5.2)

## 2 RELATED WORK

**Safety problems in Diffusion Models**   Although the GDM has achieved great success in generating synthesis content, an increasing number of safety concerns have emerged. Consequently, there has been a growing number of works trying to resolve the concerns and protect GDM from being abused. Some of them focus on removing bad concepts such as nudity, violence, or other certain concepts (Gandikota et al., 2023a; Zhang et al., 2023b; Gandikota et al., 2023b; Heng & Soh, 2023; Kumari et al., 2023a). Some of them work on protecting the property identification by adding watermarks into the diffusion model (Zhao et al., 2023; Peng et al., 2023; Cui et al., 2023). Some of them work on fairness and unbiased generation (Friedrich et al., 2023; Struppek et al., 2022). Also, some of them call attention to possible adversarial samples generated using GDM (Xue et al., 2023; Chen et al., 2023a; Liu et al., 2023; Chen et al., 2023b). With the rapid development of increasingly powerful generative models, we need to pay more attention to these safety issues.

**Protection against diffusion-based mimicry**   The most related works are some recent efforts that attempt to shield images from diffusion model-based mimicry. Photoguard (Salman et al., 2023) first proposes to raise the cost of image editing by attacking the encoder of the latent diffusion model. While it works well in image-to-image scenarios, it needs a careful redesign of the target image to be able to work under textual-inversion (Gal et al., 2022) as is reported in (Liang & Wu, 2023). Though Photoguard also provides a stronger diffusion attack, it needs to know the editing pipeline first and is too expensive to run, so in this paper, we turn to the encoder-based attack when we mention Photoguard. Similarly, Glaze (Shan et al., 2023) also proposes to attack the latent space, with more regulation terms to make the perturbation smoother. AdvDM (Liang et al., 2023) focuses on generating adversarial samples for GDMs, but it needs to calculate the expensive gradient over the denoising module. Mist (Liang & Wu, 2023) proposes to combine semantic loss with textural loss with a carefully designed target pattern, showing strong ability against different types of attacks. However, it also suffers from the heavy computational cost. Most of the previous works fail to unravel the bottleneck of the attack against the diffusion model and have not thoroughly explored the design space.

## 3 BACKGROUND

**Generative Diffusion Model**   The generative diffusion model (GDM (Song et al., 2020b; Ho et al., 2020)) is a special kind of generative model that has demonstrated superior performance. Among various kinds of GDM, the Latent Diffusion Model (LDM) (Rombach et al., 2022), a GDM in the latent space, has gained great success in text-to-image generation and image editing.

Suppose $x_0 \sim q(x_0)$ is from a real data distribution, LDM first uses an encoder $\mathcal{E}_\phi$ parameterized by $\phi$ to encode $x_0$ into latent variable: $z_0 = \mathcal{E}_\phi(x_0)$. Then, the same as other GDMs, the forward process is conducted by gradually adding Gaussian noise, generating noisy samples $[z_1, z_2, ..., z_T]$ in $T$ steps, following a Markov process formulated as $q(z_t \mid z_{t-1}) = \mathcal{N}(z_t; \sqrt{1 - \beta_t} z_{t-1}, \beta_t \mathbf{I})$. By accumulating the noise we have: $q_t(z_t \mid z_0) = \mathcal{N}(z_t; \sqrt{\bar{\alpha}_t} z_{t-1}, (1 - \bar{\alpha}_t)\mathbf{I})$, where $\beta_t$ growing from 0 to 1 are fixed values, $\alpha_t = 1 - \beta_t$, and $\bar{\alpha}_t = \Pi_{s=1}^t \alpha_s$. Finally, $z_T$ will become approximately an isotropic Gaussian random variable when $\bar{\alpha}_t \to 0$.

The reverse process $p_\theta(\hat{z}_{t-1} | \hat{z}_t)$ can generate samples from Gaussian $\hat{z}_T \sim \mathcal{N}(0, \mathbf{I})$, where $p_\theta$ can be replaced by alternatively learning a noise estimator $\epsilon_\theta(\hat{z}_t, t)$ parameterized by $\theta$. By gradually estimating the noise, we can generate $\hat{z}_0$ in the latent space: $p(\hat{z}_{0:T}) = p(\hat{z}_T) \prod_{t=1}^T p_\theta(\hat{z}_{t-1} \mid \hat{z}_t)$. Finally, $\hat{z}_0$ can be projected back to the pixel space using decoder $\mathcal{D}_\psi$ parameterized by $\psi$ as $\hat{x}_0 = \mathcal{D}_\psi(\hat{z}_0)$, and $\hat{x}_0$ are supposed to be images with high fidelity.

**Adversarial Examples for LDM**   Current works of protection against diffusion-based mimicry focus on finding an adversarial sample $x_{adv}$ given a clean image $x$, which can fool the targeted diffusion model. In order to calibrate $x_{adv}$, we use the restricted attacks (e.g. PGD (Madry et al., 2018)) widely used in adversarial sample generation. Two objective functions are widely used in existing works:

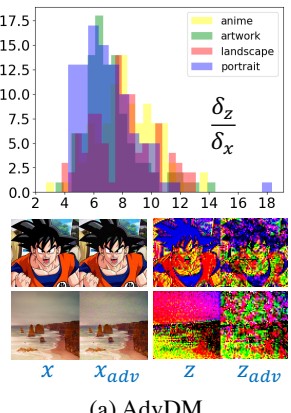
(a) AdvDM

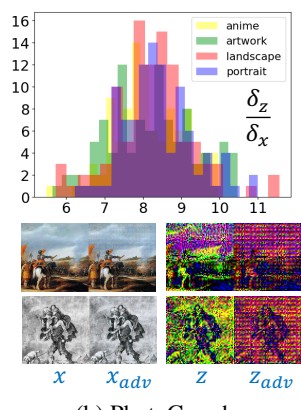
(b) PhotoGuard

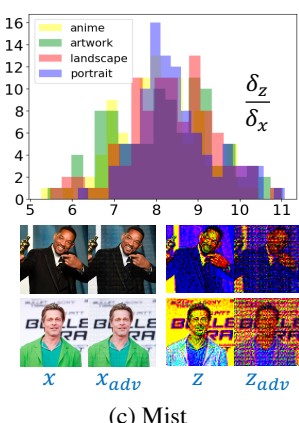
(c) Mist

Figure 2: **The $z$-Space of LDM is Vulnerable**: here we show that the $z$-space exhibit significantly greater magnitude than the $x$-space after the protection. It is a common phenomenon for current protection methods: we show statistical results on (a) AdvDM(Liang et al., 2023), (b) PhotoGuard(Salman et al., 2023) and (c) Mist(Liang & Wu, 2023). The above histogram of each method demonstrates the distribution of $\delta_z/\delta_x$ across the four domains of the dataset (anime, artwork, landscape, and portrait), where both $\delta_z$ and $\delta_x$ are computed using the $\ell_\infty$ norm and are subsequently normalized. In the lower part of each method's illustration, we provide visual representations of the original image $x$, the protected image $x_{adv}$, and their latents $z$ and $z_{adv}$ respectively.

- Semantic Loss: Liang et al. (2023) defines the semantic loss exactly as noise estimation loss during the training of an LDM, aiming to fool the denoising process, thus guiding the diffusion model to generate samples away from $q(x_0)$:

$$\mathcal{L}_S(x) = \mathbb{E}_{t,\epsilon}\mathbb{E}_{z_t \sim q_t(\mathcal{E}_\phi(x))}\|\epsilon_\theta(z_t, t) - \epsilon\|_2^2 \tag{1}$$

- Textural Loss: Salman et al. (2023); Liang & Wu (2023); Shan et al. (2023) define the textural loss by pushing the latent of $x$ towards a target latent generated by some other image $y$:

$$\mathcal{L}_T(x) = -\|\mathcal{E}_\phi(x) - \mathcal{E}_\phi(y)\|_2^2 \tag{2}$$

The final objective $\mathcal{L}_{adv}$ can be either $\mathcal{L}_S$ or $\mathcal{L}_T$ separately (Salman et al., 2023; Shan et al., 2023; Liang et al., 2023) or the combination of them (Liang & Wu, 2023). We can then run the iterations of the Projected Gradient Ascent with $\ell_\infty$ budget $\delta$ by

$$x^{t+1} = \mathcal{P}_{B_\infty(x,\delta)}\left[x^t + \eta\,\mathrm{sign}\nabla_{x^t}\mathcal{L}_{adv}(x^t)\right] \tag{3}$$

where $\mathcal{P}_{B_\infty(x,\delta)}(\cdot)$ is the projection operator on the $\ell_\infty$ ball. Note we use superscript $x^t$ to represent the iterations of the PGD, while subscript $x_t$ to represent the diffusion steps.

While it may seem intuitive that the gradient steers the sample $x^t$ to deceive the LDM, a deeper examination of the underlying mechanics is needed. In the subsequent section, we show a surprising finding: while optimizing $\mathcal{L}_S$ with gradient ascent is effective against the LDM, the actual improvement comes from attacking the encoder of the LDM.

## 4 THE BOTTLENECK OF ATTACKING AN LDM

The protection against diffusion-based mimicry is one type of attack against the LDMs. The current prototype attacks the LDMs using gradient-based methods by maximizing the semantic loss in Eq.1, aiming to mislead the denoising process in the LDM.

Here we present a comprehensive study of the attacks based on semantic loss, showing that what is really attacked is the encoder $\mathcal{E}_\phi$. Meanwhile, we show that the denoising module $\epsilon_\theta$ is much more robust than $\mathcal{E}_\phi$. We support these conclusions by providing the following evidence:

- The latent space has a **much larger attack budget** than pixel space, and more perturbations are injected into the latent space during an attack.

Figure 3: **Perturbations in Latent Space Reflect the Editing Results:** when $x_{adv}$ is generated, we have $\text{Edit}_{\phi,\theta,\psi}(x_{adv}, t)$ highly reflected by $\mathcal{D}_\psi(E_\phi(x_{adv}))$: sharing similarly unrealistic patterns such as bluring, colorful pattern or target pattern. This further proves that the changes in the $z$-space dominate the attack.

- The edited results over protected images are **highly correlated with** the perturbation in the latent space, showing that the denoiser is not the key factor.
- It often fails to run attacks in the latent space, showing that the **denoiser $\epsilon_\theta$ is robust.**

## 4.1 THE ATTACK IN $z$-SPACE HAS MUCH HIGHER BUDGET

We first show that during the attack, the latent representation is dramatically changed, though the perturbations are minor in the pixel space. We refer to $z$-space as the space of the encoded images: $\{z|z = \mathcal{E}_\phi(x), x \sim q(x_0)\}$. Since we use project gradient ascent over the semantic loss with a fixed $\ell_\infty$ budget $\delta_x$ in the $x$-space, the perturbation $|x_{adv} - x|_\infty \leq \delta_x$ is restricted. Similarly, we have the perturbations in the $z$-space: $\delta_z = |z_{adv} - z|_\infty$. We *normalized* the two spaces for comparison.

While $\delta_x$ is always fixed, we want to show that perturbations in the $z$-space have much larger budgets than that in the $x$-space, which means that the latent representation changes significantly during the attack, namely, $\delta_z/\delta_x \gg 1$.

From Figure 2 we clearly see that: for all three optimization-based protection methods, the latent in the $z$-space always has dramatic changes after the attack. In contrast, the perturbations in $x$-space are strictly bounded. Numerically, the attack in the $z$-space can be 10 times larger, implying the fact that the encoder is quite vulnerable to attack.

Although we aim to attack the entire LDM, including the encoder module and denoiser module, the gradient follows a shortcut: attacking the $z$-space is much easier. While this finding can show that the encoder $\mathcal{E}_\phi$ is vulnerable, we still need more clues to safely land on the conclusion that the denoiser $\epsilon_\theta$ is robust to be attacked, which are further explored in Section 4.3.

## 4.2 PERTURBATIONS IN $z$-SPACE REFLECTS THE EDITING RESULTS

Next, we present another clue to show that the perturbations in the $z$-space dominate the editing results, reflecting that the denoiser is barely attacked. Defining $\text{Edit}_{\phi,\theta}(x, t)$ as the SDEdit (Meng et al., 2021) procedure to edit an image, where $t$ measures how strong the edit is applied. For protection, we hope the edited results $\text{Edit}_{\phi,\theta,\psi}(x_{adv}, t)$ to be messy and unrealistic. Different protection methods show different unrealistic patterns: AdvDM (Liang et al., 2023) tends to make the editing results tortured and colorful, Mist (Liang & Wu, 2023) and PhotoGuard (Salman et al., 2023) tend to make the editing results similar to the target image pattern, and AdvDM(-) with gradient descent (will be introduced in Section 5.2) is prone to blur the edited images.

In Figure 3 we show that for all the methods mentioned above, $\text{Edit}_{\phi,\theta,\psi}(x_{adv}, t)$ is highly reflected by $\mathcal{D}_\psi(E_\phi(x_{adv}))$ where denoiser is not involved. This phenomenon also help us understand that the perturbations against the encoder guides take the main part of the attack.

## 4.3 THE DENOISER MODULE IS MUCH MORE ROBUST

We push it forward by directly attacking the latent space via a modified Eq 4 as

$$z^{t+1} = \mathcal{P}_{B_\infty(z,\delta)}\left[z^t + \eta \operatorname{sign}\nabla_{z^t}\mathcal{L}_{adv}(z^t)\right] \tag{4}$$

$z_{raw}$           $z_{adv}$ ($\ell_\infty \approx 0.5$)  Can still get good $\hat{x}_0$ at each timestep, attack failed

Figure 4: **Directly Attacking the Latent Space Does not Work:** here we show attacks in the latent space with $\ell_\infty$ budget of 0.5 (normalized, nearly 10-times larger budget as in $x$-space), running PGD attacks by sampling timestep $t$, we find that after the attack, the predicted noise is still reasonable, which means that the attack did not fool the denoiser that much.

where $\mathcal{L}_{adv}(z^t)$ is still defined as the loss of noise estimation in LDM. From Figure 4 we can see that, though we set the budget to be much larger than that in the pixel space, the direct attacks in the $z$-space cannot effectively deceive the denoiser. In conclusion, it is hard to fool the denoiser by adding restricted small perturbations, which may be due to the stochastic inputs of the denoiser module. In contrast, the encoder is shown to be vulnerable to adversarial attacks: we can add small perturbations to the original image to make the decoded image messy. We include more results in Section C in the appendix to further support this argument.

## 5 APPROACHES AND METHODOLOGY

### 5.1 FASTER SEMANTIC LOSS WITH SCORE DISTILLATION

Since the bottleneck for attacking against the LDM is the encoder, it is unnecessary to allocate too much computational effort to calculate the gradient of the denoise module, which is expensive.

The semantic loss $\mathcal{L}_S$ is introduced as the expectation over the error of noise estimation at each time step. Nevertheless, a noteworthy challenge is obvious when examining Eq. 1 and Eq. 4: the computation of the term $\nabla_x \mathcal{L}_S(x)$ proves to be computationally intensive, particularly when we are required to perform more than 100 iterations for the update process. Another concern is the substantial GPU memory usage, which places a significant burden on individual users. To resolve these concerns, we turn to an approximation

$$\nabla_x \mathcal{L}_S(x) = \mathbb{E}_{t,\epsilon}\mathbb{E}_{z_t}\left[\lambda(t)(\epsilon_\theta(z_t,t)-\epsilon)\frac{\partial \epsilon_\theta(z_t,t)}{\partial z_t}\frac{\partial z_t}{\partial x_t}\right] \approx \mathbb{E}_{t,\epsilon}\mathbb{E}_{z_t}\left[\lambda(t)(\epsilon_\theta(z_t,t)-\epsilon)\frac{\partial z_t}{\partial x_t}\right] \quad (5)$$

The above equation reflects the idea of Score Distillation Sampling in (Poole et al., 2022). Here we note $\nabla_x \mathcal{L}_{\text{SDS}}(x) = \mathbb{E}_{t,\epsilon}\mathbb{E}_{z_t}\left[\lambda(t)(\epsilon_\theta(z_t,t)-\epsilon)\frac{\partial z_t}{\partial x_t}\right]$.

We refer to the above gradient update as the SDS version of the adversarial attacks against LDM, which can be a plug-and-play for all the previous methods using semantic loss. Using the SDS version can dramatically make the calculation of the semantic loss cheaper, both from the viewpoint of time consumption and GPU memory occupation. Moreover, Poole et al. (2022) shows that the Jacobian $\frac{\partial \epsilon_\theta(z_t,t)}{\partial z_t}$ is unstable to calculate and poorly conditioned for small noise levels. We will further demonstrate that this ingredient can not only speed up the protection but also make the protection even better (Table 1, 2).

### 5.2 GRADIENT DESCENT OVER SEMANTIC LOSS MAKES GOOD PROTECTION

Previous methods show that maximizing the semantic loss can make the attacked images fool the editor into generating unrealistic patterns, while the perturbation itself always turns to largely affect the original images, making the perturbation not natural. Here we provide a surprising finding: minimizing the semantic loss can also achieve good attacks and show more natural perturbations than maximizing the semantic loss.

Specifically, we reverse the optimization objective $\mathcal{L}_S$ by following gradient descent, that is, minimizing the semantic loss. Intuitively, it will guide the LDM to make better predictions. However, through experiments, we find that it will actually blur the edited results, which is also one type of protection. Moreover, we found that the perturbations added by minimizing the semantic loss are more harmonious with the original images, showing similar edge patterns.

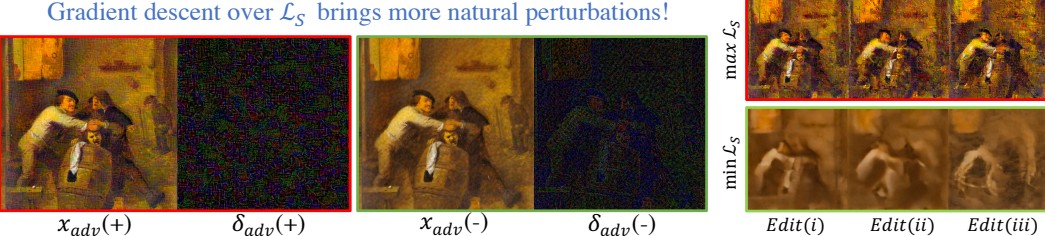

Figure 5: **Minimizing Semantic Loss Brings More Natural Protection**: [Left] We show the attacked images $x_{adv}(+)$ using gradient ascent (red boundary), $x_{adv}(-)$ with gradient descent (green boundary), and their perturbations $\delta_{adv}(+)$, $\delta_{adv}(-)$. [Right] The SDEdit results over the two kinds of protected images, with increasing strength Edit($i, ii, iii$). Zoom in on a computer screen for better visualization.

| Protection Method | SSIM↑ | PSNR↑ | LPIPS↓ | VRAM↓ | TIME↓ | P-Speed↑ | HumanEval↑ |
|---|---|---|---|---|---|---|---|
| AdvDM | 0.714 | 29.074 | 0.437 | ~16G | ~65s | 0.05 | 2.94 |
| MIST | 0.689 | 28.897 | 0.453 | ~16G | ~65s | 0.05 | 2.44 |
| PhotoGuard | 0.684 | 29.0 | 0.456 | ~8G | ~30s | 0.25 | 2.27 |
| AdvDM(-) | 0.677 | 28.844 | 0.445 | ~16G | ~65s | 0.05 | - |
| SDS(+) | **0.719** | 29.413 | 0.426 | ~8G | ~30s | 0.25 | - |
| SDS(-) | 0.698 | **29.562** | **0.425** | ~8G | ~30s | 0.25 | **4.55** |
| SDST($\lambda = 5$) | 0.699 | 29.288 | 0.439 | ~10G | ~45s | 0.11 | 2.75 |

Table 1: **Quantiative Results of Perturbations Generated by Different Protection Methods**

In Figure 5 we illustrate the effect of applying gradient descent (GD) over the semantic loss compared with gradient ascent (GA). We observe that:

1. The perturbations generated using GD exhibit a more natural and harmonious appearance than GA. GD's optimization process closely aligns with the underlying structure of the original image.

2. GD-based protection tends to better eliminate the information from edited images by blurring it, while GA-based protections try to bring in more chaotic patterns.

Combined with the SDS acceleration we presented in the previous section, we propose some novel protection strategies named SDS(+), SDS(-), and SDST, where SDS means that Score Distillation Sampling is applied, (+) and (-) refer to the two strategies (descent and ascent) regarding the semantic loss, and SDST means textual loss is also used. More detailed descriptions of each method are put in Table 3 in the appendix.

## 6 EXPERIMENTS

We demonstrate the performance of all the methods within the design space. Some of these methods have been previously proposed, while others have been newly constructed using our novel strategies. By presenting comprehensive experimental results in quantitative and qualitative aspects, we aim to answer the following questions:

- **(Q1)**: Are SDS versions of the protections still effective compared with the original version?
- **(Q2)**: Is gradient descent over semantic loss better than gradient ascent against mimicry?
- **(Q3)**: What are the pros and cons of all the methods working on different protection tasks?

**Models and Datasets** We work on the pre-trained LDM provided in (Rombach et al., 2022) as our backbone model, which is the mainstream model used in AI-based mimicry (Liang et al., 2023). For evaluation datasets, while the previous works either focus more on portraits or artworks, we collect four small subsets including anime, artworks, landscape, and portraits. We collect the anime and portrait data from the internet, the landscape data from (Arnaud, 2020), and the artworks subset from WikiArt (Nichol, 2016). Details about the dataset are shown in the appendix.

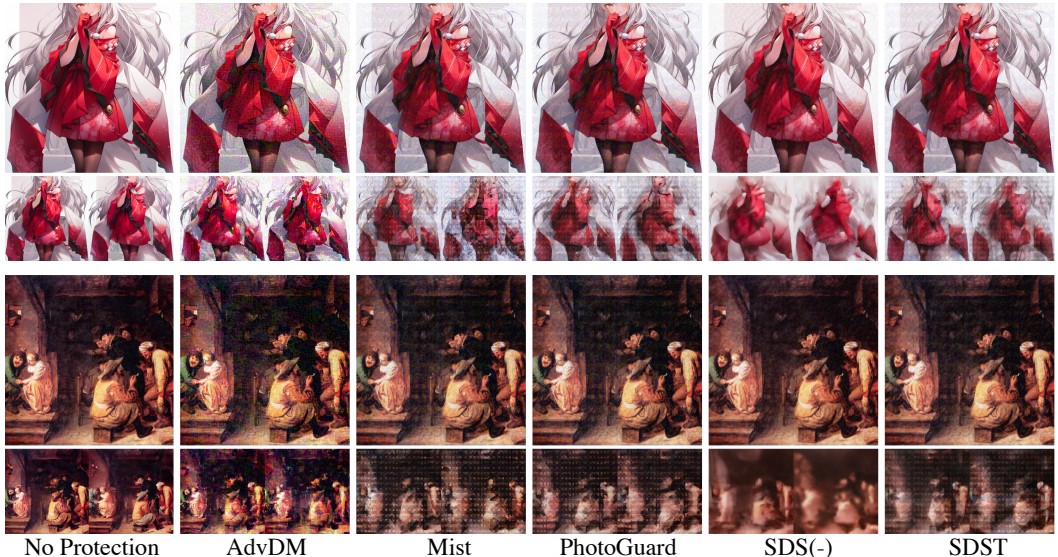

No Protection    AdvDM    Mist    PhotoGuard    SDS(-)    SDST

Figure 6: **Results of Protection Against SDEdit**: each column represents one protection method (including no protection), the two smaller figures below each protected image are generated using SDEdit, with two different strengths (the left one is smaller than the right one).

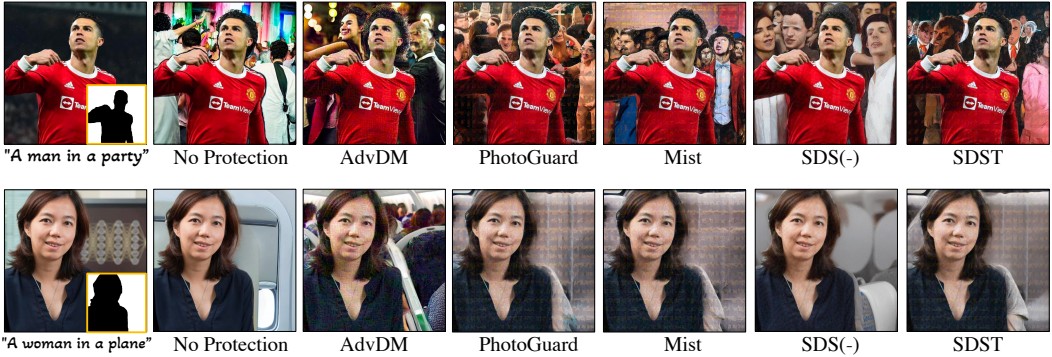

Figure 7: **Results of Protection Against Inpainting**: the first column shows the clean images to be inpainted, with given masks and prompts (unknown to the defenders), and then the left columns show inpainting results of different protection approaches.

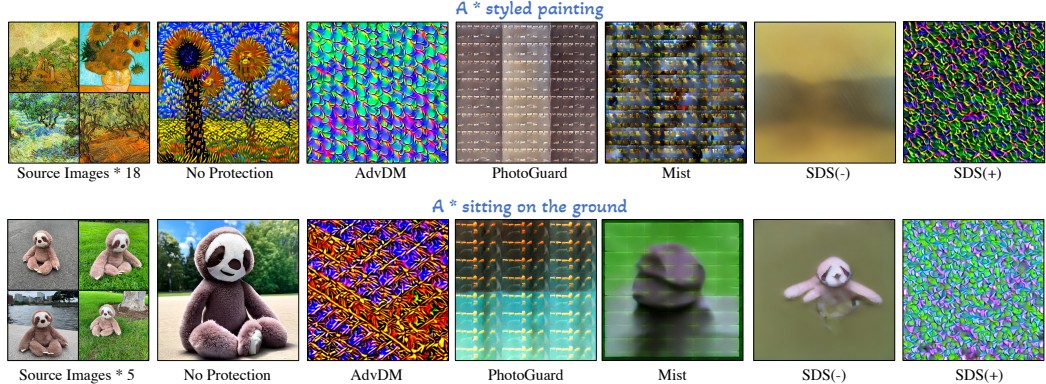

Figure 8: **Results of Protection Against Textual Inversion**: the first column is the subset of clean images we used to train the embedding "*", then we show the generated images with embedding trained on images protected using different protection approaches.

| Methods | FID-score↑ | | | IA-score↓ | | | LPIPS↑ | | | PSNR↓ | | | HumanEval↑ |
|---|---|---|---|---|---|---|---|---|---|---|---|---|---|
| Edit Strength | i | ii | iii | i | ii | iii | i | ii | iii | i | ii | iii | - |
| Clean | 62.36 | 72.68 | 83.87 | 0.956 | 0.943 | 0.925 | 0.092 | 0.128 | 0.165 | 31.456 | 30.782 | 30.285 | 1.16 |
| AdvDM | 251.82 | 254.32 | 297.33 | 0.771 | 0.738 | 0.706 | 0.511 | 0.571 | 0.632 | 28.986 | 28.791 | 28.650 | 2.03 |
| MIST | 346.70 | 367.40 | 372.00 | **0.648** | 0.616 | 0.587 | 0.543 | 0.564 | 0.581 | 28.604 | 28.486 | 28.367 | 3.19 |
| PhotoGuard | 342.48 | **369.21** | **375.32** | 0.652 | 0.613 | 0.587 | 0.540 | 0.561 | 0.578 | 28.610 | 28.479 | 28.365 | 3.23 |
| AdvDM(-) | 199.45 | 211.03 | 221.13 | 0.738 | 0.704 | 0.671 | **0.604** | **0.649** | **0.692** | 28.733 | 28.632 | 28.522 | - |
| SDS(+) | 242.53 | 256.84 | 299.25 | 0.782 | 0.751 | 0.719 | 0.511 | 0.559 | 0.616 | 29.001 | 28.812 | 28.675 | - |
| SDS(-) | 206.01 | 220.15 | 231.57 | 0.714 | 0.677 | 0.644 | 0.535 | 0.583 | 0.632 | 28.704 | 28.602 | 28.503 | 4.34 |
| SDST($\lambda = 1$) | **346.75** | 356.84 | 365.04 | 0.649 | **0.612** | 0.589 | 0.535 | 0.557 | 0.576 | **28.600** | **28.478** | **28.354** | - |
| SDST($\lambda = 5$) | 294.92 | 318.15 | 322.80 | 0.670 | 0.632 | **0.577** | 0.480 | 0.518 | 0.552 | 28.682 | 28.525 | 28.413 | 3.56 |

Table 2: **Quantiative Measurement of Different Protections against SDEdit.**

**Baseline Methods and Metrics** We compare our methods with three main-stream open-sourced protection methods: including AdvDM (Liang et al., 2023), PhotoGuard (Salman et al., 2023) ($\mathcal{L}_T$ only) and Mist (Liang & Wu, 2023). The two main strategies (SDS and GD) we proposed in the previous sections combine with each other to form our new proposed methods, forming:

AdvDM(-)=[AdvDM+GD]; SDS(+)=[AdvDM+SDS]; SDS(-)=[SDS+GD]; SDST=[SDS+GD+$\mathcal{L}_T$]

each of which can be regarded as the previous strategy with our new plug-and-play strategies. For the textural loss $\mathcal{L}_T$, we use the most effective target image as was reported in the Mist paper for all the methods. $\lambda$ is used as a scaling factor for the textural loss. A detailed summarization of all the methods in the design space is put in Table 3 in the appendix. As for the quantitative metrics, we give a detailed demonstration in Section B.4 in the appendix.

**Threat Model** We consider the following three mimicry scenarios: (1) Basic SDEdit: it is the cheapest and easiest edition a mimicker can do over a single image, which also serves as a more fundamental baseline task to measure a given protection (2) Inpainting: a more flexible mimicry and widely used nowadays since the masked part are unknown during the attack, it turns out to be more challenging. (3) Textual Inversion (TI): different from the previous two scenarios where the objective function of our attacks directly works on the LDM, the mimicker with TI can learn a special token "*" trained on a subset of images of a single object or style, then they can use prompts like "a photo a * sitting on grass" to generate new synthesis.

**Protecting Results** Combined with all the experimental results, we answer all the questions. **(Q1):** Applying SDS can largely save the computational resources by 50% (Table 1) without losing the effectiveness (Table 2). That means SDS turns out to be a free lunch, which can be alternatively used in the semantic loss to attack the LDM. **(Q2):** Through experiments we found that gradient descent can achieve a strong protection with more natural perturbations. At the same time, in Figure 6 we can see that SDS(-) shows strong protection by the blurring effect on the edited image. It also shows to be preferred in human evaluations in Table 1 and Table 2. **(Q3):** Figure 7, 8 show results of protection against inpainting and textual inversion, from which we can see that: all the protection can fool the LDM-based inpainting, making the inpainted results unrealistic, where the perturbation of SDS(-) is more natural. For textual inversion, we find that SDS(+) shows the strongest protection as AdvDM does, while it is much cheaper to run.

# 7 CONCLUSIONS

In this paper, we propose enhanced protection against diffusion-based mimicry by pointing out the actual bottleneck when attacking the LDM. We present intriguing findings, revealing that the primary target of attacks is the encoder, while the denoiser remains robust against adversarial attacks. Furthermore, we demonstrate that applying gradient descent on semantic loss can also provide protection with a more natural perturbation style. Then we propose a more effective protection framework by introducing SDS as a free lunch. Finally, we demonstrate through extensive experiments to support our findings and show the effectiveness of our proposed methods. Our work also has its limitations, it focuses only on the latent diffusion model, and exciting future directions may be design effective attacks against the diffusion model in the pixel space.

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

CONTENTS

# Appendix

In the appendix part, first, we will offer more details on the algorithm in Section A, then we show the details about our experiment settings, including dataset and metrics in Section B. After that, we will put more experimental results in Section C. We also conduct experiments purifying the protections using some popular purification methods in Section D. In Section E, we push the protection forward to black box settings by transferring protection generated on one LDM to other LDMs. Moreover, we provide more explanation for SDS approximation in Section G, F and why we can minimize the semantic loss in Section H. Finally, we show the settings of our human evaluation survey in Section I.

## A  DETAILS ABOUT ALGORITHMS

We provide a PyTorch-styled pseudo code to show how to attack the latent diffusion model (LDM) with SDS acceleration, and how does the gradient descent work:

```python
import torch
ldm = load_latent_diffusion_model()
x   = load_clean_image()
# start optimization
for _ in range(iterations):
    x = x.detach().clone()
    x.requires_grad=True
    z = encoder(x)
    noise = sample_std_gaussian()
     # forward diffusion process
    z_t = q_sample(z, noise)
    # SDS gradient, only inference
    with torch.no_grad():
        # SDS gradient in z-space
        sds_grad = ldm(z_t, t) - noise
    z.backward(gradient=sds_grad)
    grad = x.grad().detach() # final gradient in x-space
    # projected gradient descent/ascent
    if mode == 'gradient ascent':
        x = x + grad.sign() * step_size
    elif mode == 'gradient descent':
        x = x - grad.sign() * step_size
    # clip to budget restriction
    x = clip(x, eps)

x_adv = x
# run down streaming mimicry with protected image
run(x_adv, task)
```

from the above code, we can see that the gradient information of the denoiser does not need to be saved during the protection, which makes it much faster and also saves a lot of GPU memory. This enables individual users to run the protection algorithm more easily.

Also, we have a variety of new proposed protection methods under our design space: we summarize the design space as { semantic loss ($\mathcal{L}_S$), textural loss ($\mathcal{L}_T$), SDS, gradient descent (GD), gradient ascent (GA) }. All the methods evaluated in this paper can be constructed in the design space, and here we summarize all the methods as follows in Table 3.

## B  DETAILS ABOUT OUR EXPERIMENTS

### B.1  DATASET

While the previous works either focus more on portraits or artworks, we collect four small subsets including anime, artworks, landscape, and portraits. We collect the anime and portrait data from the

| Methods | Component | Perturbation | Consumption | SDEdit | Inpainting | Textual Inversion |
|---------|-----------|--------------|-------------|--------|------------|-------------------|
| AdvDM | $\mathcal{L}_S$+ GA | * | * | ** | ** | *** |
| Mist | $\mathcal{L}_S$+ GA + $\mathcal{L}_T$ | ** | * | ** | ** | ** |
| PhotoGuard | $\mathcal{L}_T$ | ** | *** | * | ** | ** |
| AdvDM(-) | $\mathcal{L}_S$+ GD | *** | * | ** | ** | * |
| SDS(+) | $\mathcal{L}_S$+ GA+ SDS | * | *** | ** | ** | *** |
| SDS(-) | $\mathcal{L}_S$+ GD+ SDS | *** | *** | *** | ** | * |
| SDST | $\mathcal{L}_S$+ GD+ SDS + $\mathcal{L}_T$ | *** | ** | ** | ** | *** |

Table 3: **Summary of All the Protection Methods in our Design Space**: we summarize all the protection methods we currently have, and all can be composed into some components in the design space we proposed. The first three rows include methods that are proposed in previous works, and the left four rows include the new protection methods first proposed in our paper, with new strategies SDS and GD marked in red. We show the strength of all these methods from the perspective of the quality of perturbation (whether it is natural), the computational consumption, and their performance on SDEdit, Inpainting, and Textual Inversion respectively. We use stars to measure them roughly, more stars represent better performance (e.g. more natural perturbations, less consumption, better protection).

internet, the landscape data from (Arnaud, 2020), and the artworks subset from WikiArt (Nichol, 2016). The size of the dataset is 100 for anime and portrait subsets and 200 for landscape and artwork subsets. Samples of the dataset can be found in Figure 9. For the inpainting task, we use the portrait subset in our dataset, using Grounded-SAM to get the mask of the human object. For the textual inversion task, we use samples from the dataset provided by (Ruiz et al., 2023).

## B.2 Implementation of Baselines

For AdvDM and Mist, we follow the settings in the original paper. For PhotoGuard, we use the pattern proposed in Mist as the target image, which is shown to be the most effective pattern. All the input images have a resolution of $512 * 512$.

For all the methods, we use $\delta = 16/255$ as the $\ell_\infty$ budget, $\alpha = 1/255$ as the step size and run $100$ iterations in the format of PGD attacks.

## B.3 Implementation of the Threat Model

All the threat model experiments in this paper can be run on one single A6000 GPU without parallelization.

For the global SDEdit, we use DDIM (Song et al., 2020a) to accelerate the reverse sampling, setting the total respaced timestep to be 100, in Figure 6, we show the SDEdit results of forward strength 0.2 and 0.3. The text prompts are set to 'a anime picture, 'a landscape picture', 'a artwork painting' and 'a portrait photo' for each subset.

For image inpainting, we use the StableDiffusion Inpainting pipeline provided by Diffusers: https://huggingface.co/docs/diffusers/using-diffusers/inpaint, using the default settings in the pipeline, with strength set to 1.0 and text-guidance set to 7.5.

For textual inversion, we also use the pipeline provided in Diffusers: https://huggingface.co/docs/diffusers/training/text_inversion, where we set the learning rate to $5 * 10^{-4}$ and train the embedding for 2000 iterations.

## B.4 Metrics

Here we introduce the quantitative measurement we used in our experiments: (1) To measure the quality of naturalness and imperceptibility of the generated perturbations, we use Fréchet Inception Distance (FID) (Heusel et al., 2017) over the collected dataset, Structural Similarity (SSIM) (Wang et al., 2004) and Perceptual Similarity (LPIPS) (Zhang et al., 2018) compared with the original image. Also, we compare the speed of protection, using metrics including the VRAM occupation (VRAM), time consumption (TIME) and the parallel speed (P-Speed, image generated per

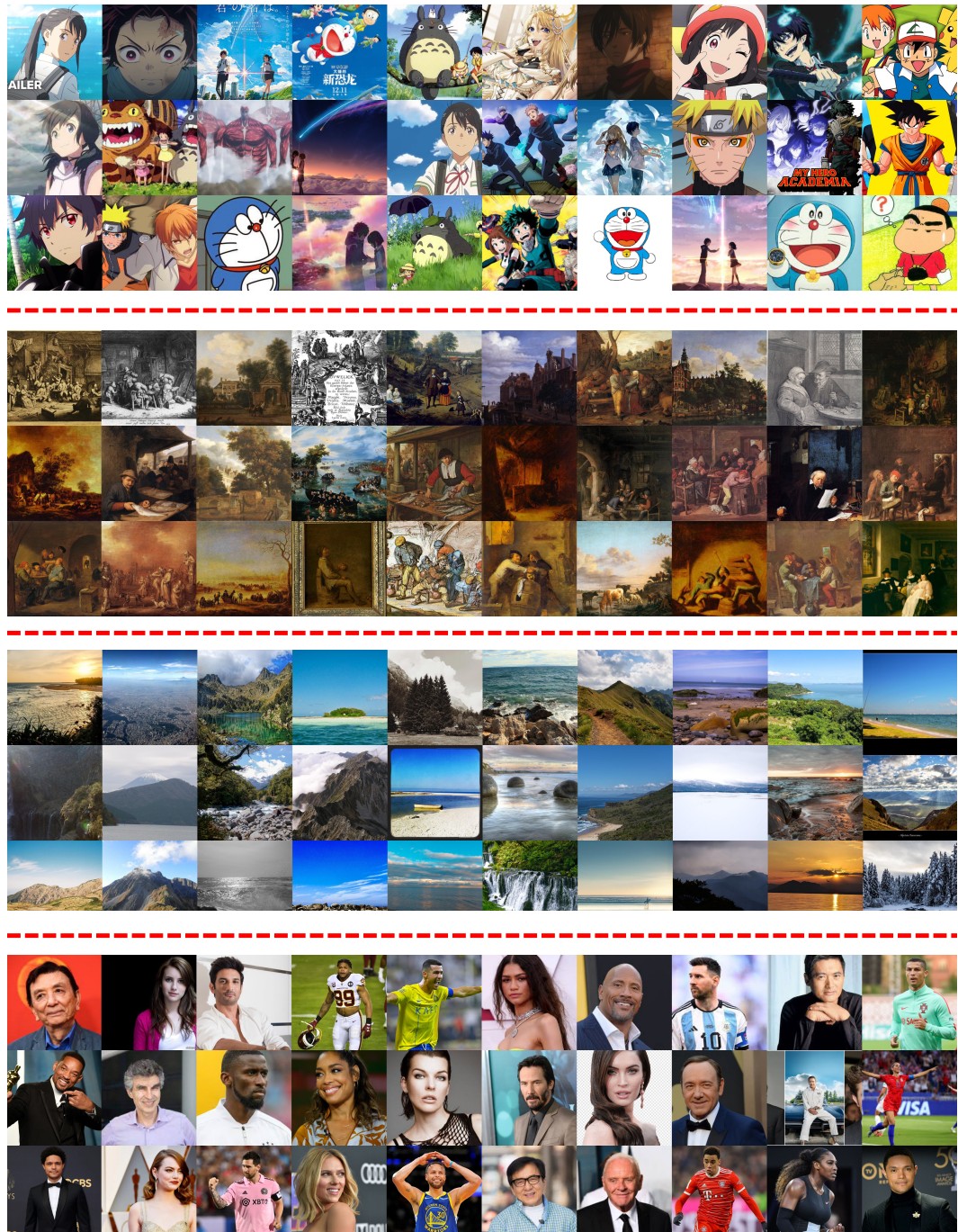

Figure 9: **Examples of Our Dataset**: in each part divided using red dotted lines, we show some samples from the subset of anime, artworks, landscape, and portraits respectively. We want to cover more kinds of mimicry scenarios to evaluate the performance of each protection method.

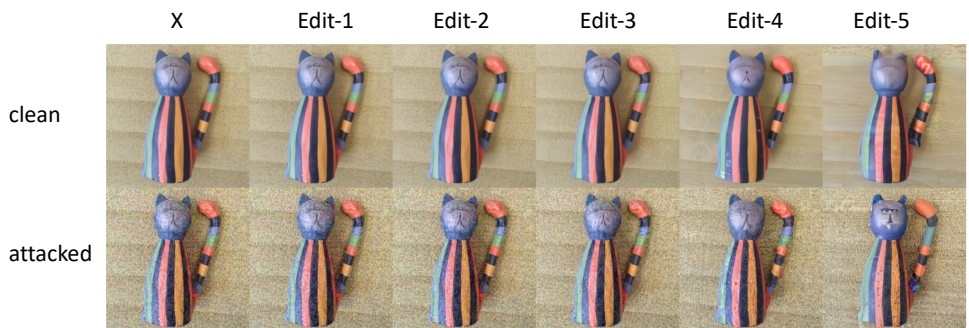

Figure 10: **Attacking a Pixel-based Diffusion Model** we attack the `https://github.com/openai/guided-diffusion` with $\delta = 16/255$ and $n = 100$, from which we can see the attack failed.

second per G of VRAM). (2) To measure the protection results, we use FID, LPIPS, Peak Signal-to-Noise Ratio (PSNR) (Huffman, 1952), and Image-Alignment Score (IA-score) (Kumari et al., 2023b) which calculated the cosine-similarity between the CLIP embedding of the protected image and the original image. Also, we have human evaluations which are collected using surveys, which is a more convincing way to evaluate the quality of protections, more settings can be found in the appendix.

## C  MORE EXPERIMENTAL RESULTS

We also provide more supplementary results of our experiments. In Figure 12 and Figrue 13, we show more visualization of the SDEdit results of different protections, from which we can see that GD brings more natural perturbations than other methods and also show effective protections.

We also show results for inpainting in Figure 14, which is a more challenging task than SDEdit, since the mask is unknown during the attack. It turns out that, all the methods can effectively make the inpainted image unrealistic in different styles.

We also show more results to support our claim that the denoiser is quite robust, we directly attack the denoiser of the LDM using three different budgets: $\delta = 16, 32, 256$. In Figure 11 we show more results, which can be used to further prove that the denoiser itself is quite robust to adversarial attacks.

We also show results of attacking a pixel-based diffusion model without encoder-decoder structure and we find that the current gradient-based attacks cannot work, showing that the denoiser is actually quite robust in Figure 10.

## D  AGAINST DEFENDING METHODS

We also provide results of SDS(-), AdvDM, Mist and PhotoGuard under some famous defense methods including Adv-Clean, Crop & Resize and JPEG compression.

- Adv-Clean: `https://github.com/lllyasviel/AdverseCleaner`, a training-free filter-based method that can remove adversarial noise for a diffusion model, it works well to remove high-frequency noise
- Crop & Resize: we first crop the image by $20\%$ and then resize the image to the original size, it turns out to be one of the most effective defense methods (Liang & Wu, 2023).

Z-space attack: $\delta = 16$

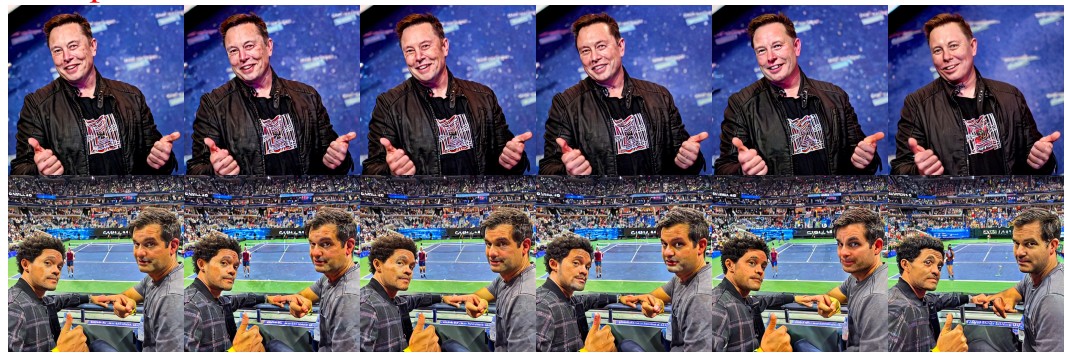

Z-space attack: $\delta = 32$

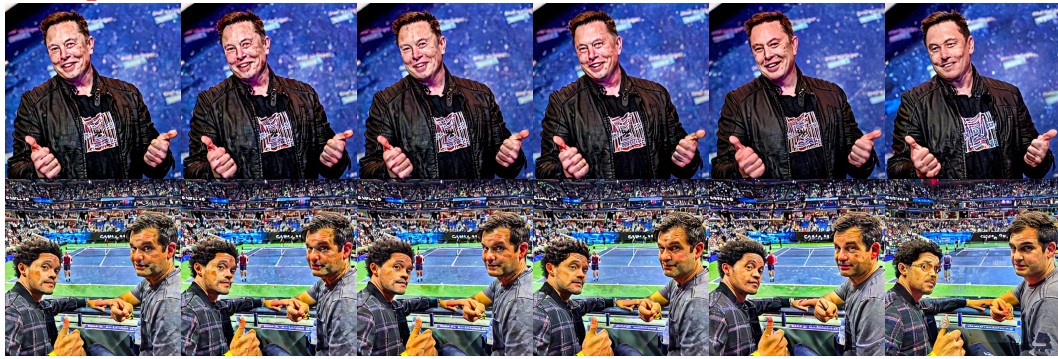

Z-space attack: $\delta = 256$

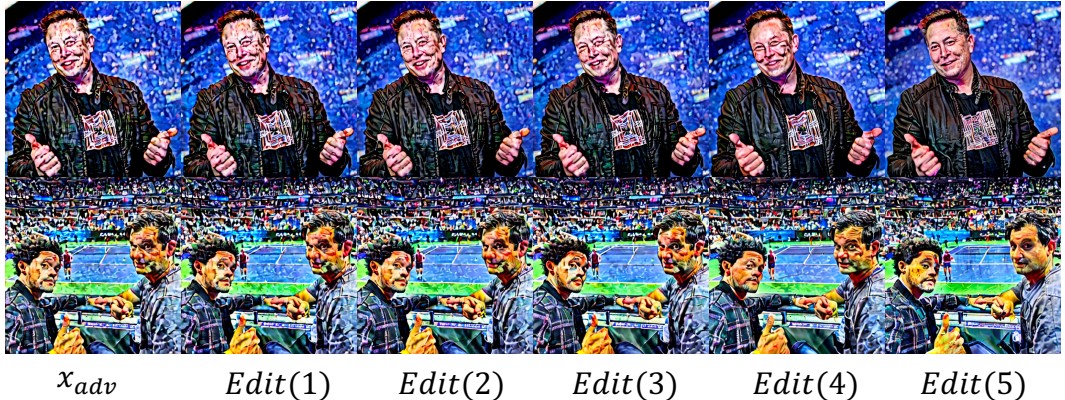

$x_{adv}$    $Edit(1)$    $Edit(2)$    $Edit(3)$    $Edit(4)$    $Edit(5)$

Figure 11: **Directly Attacking $z$-space:** we conduct experiments on three different budgets: $\delta = 16, 32, 256$ when directly attacking the latent representation in LDM. The first column is the attacked $z$-space latent projected back to $x$-space, and the following columns are results after SDEdit with an increasing editing strength. From the figure we can find that this kind of attack fails to work , the images after SDEdit still preserve better similarity as the attacked image, even when the budgets are getting as large as 256.

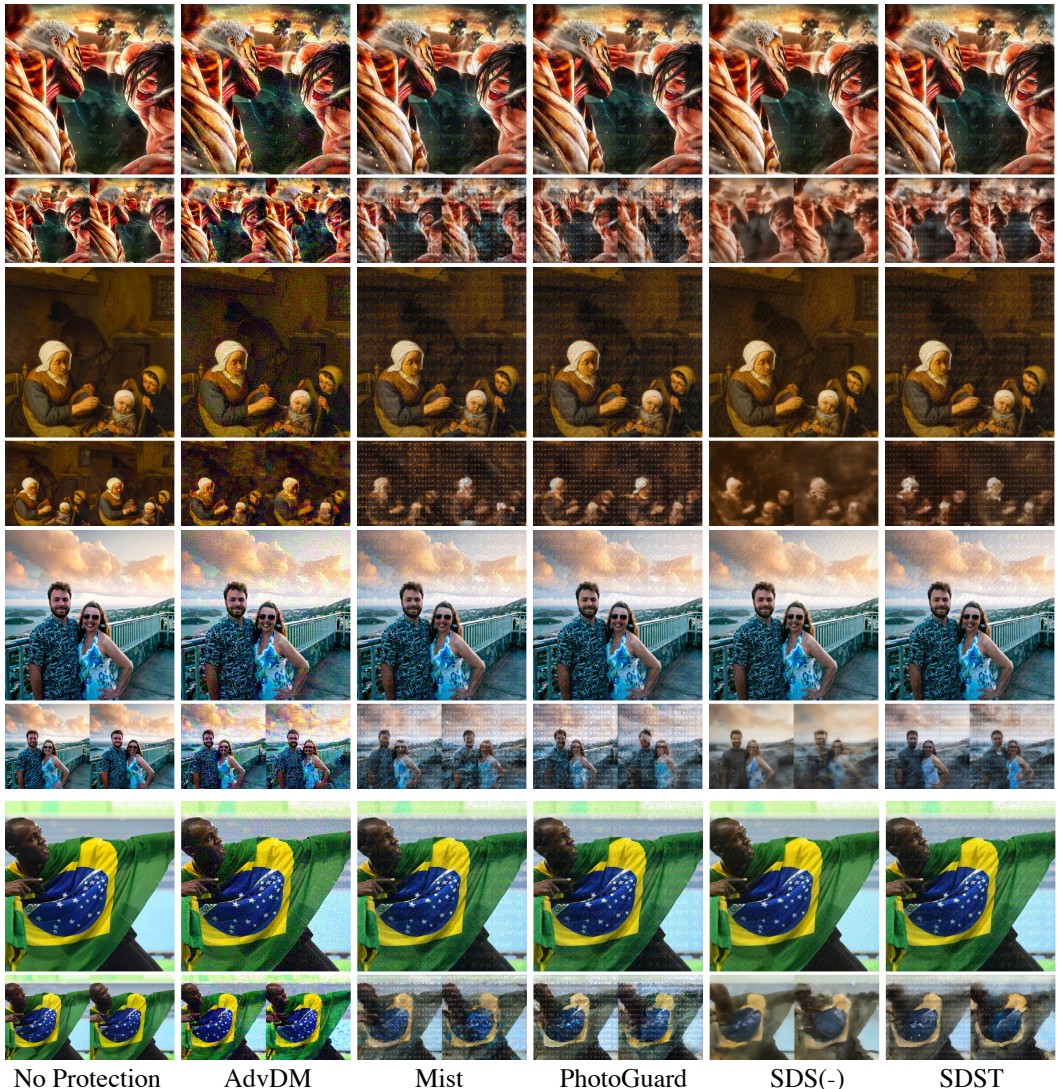

Figure 12: **More Results of Protection Against SDEdit (1/2)**: each column represents one protection method (including no protection), the two smaller figures below each protected image are generated using SDEdit, with two different strengths (the left one is smaller than the right one).

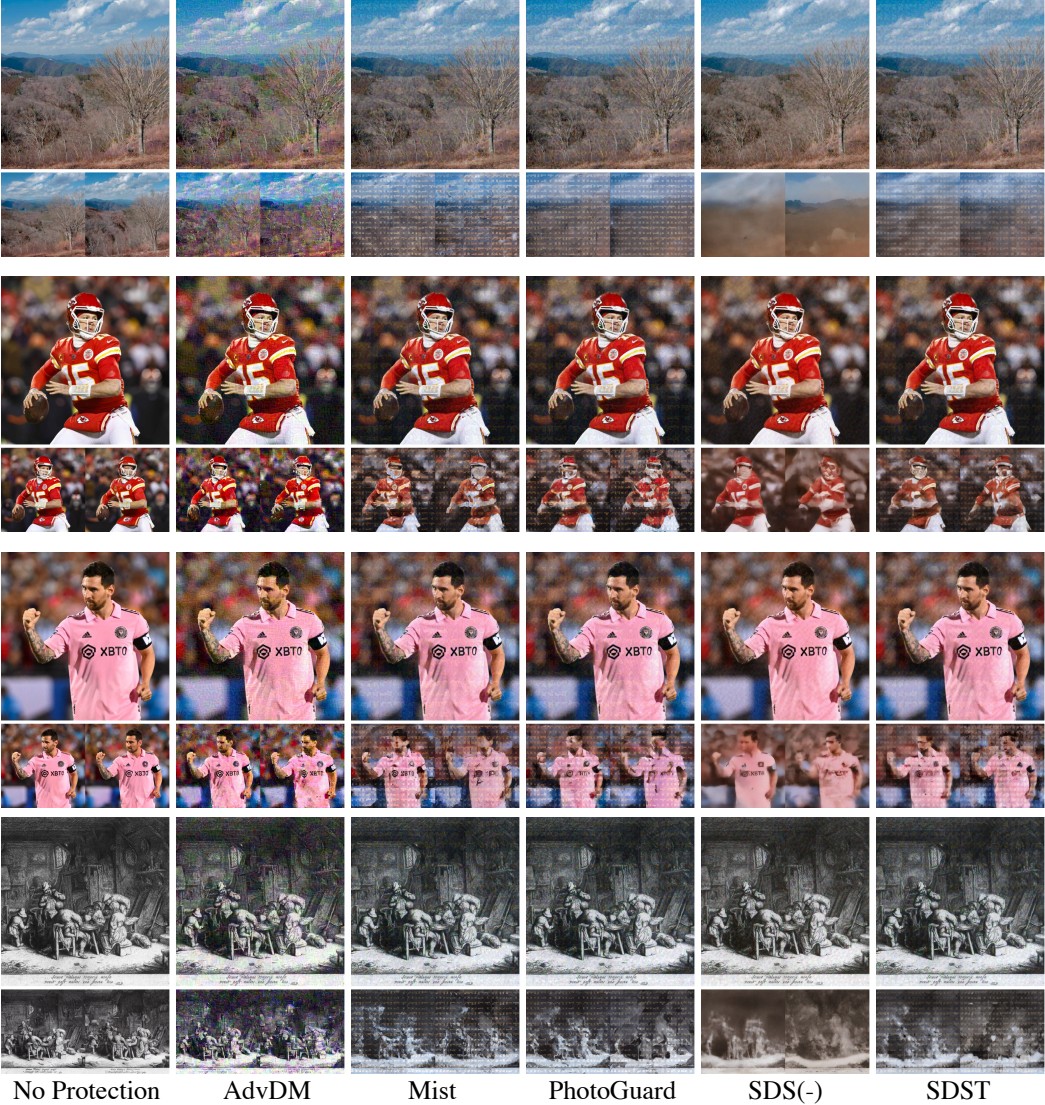

Figure 13: **More Results of Protection Against SDEdit (2/2)**: each column represents one protection method (including no protection), the two smaller figures below each protected image are generated using SDEdit, with two different strengths (the left one is smaller than the right one).

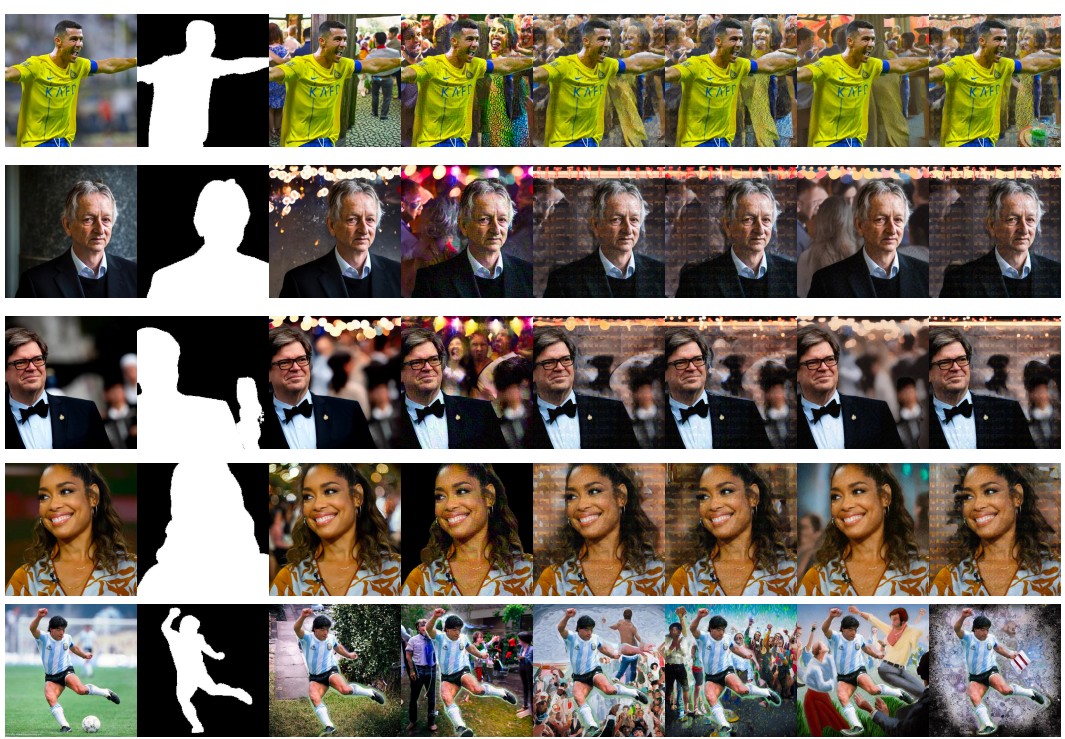

Figure 14: **More Results of Protection Against Inpainting**: From left to right: clean image, mask, clean inpainting, AdvDM, Mist, PhotoGuard, SDS(-), and SDST.

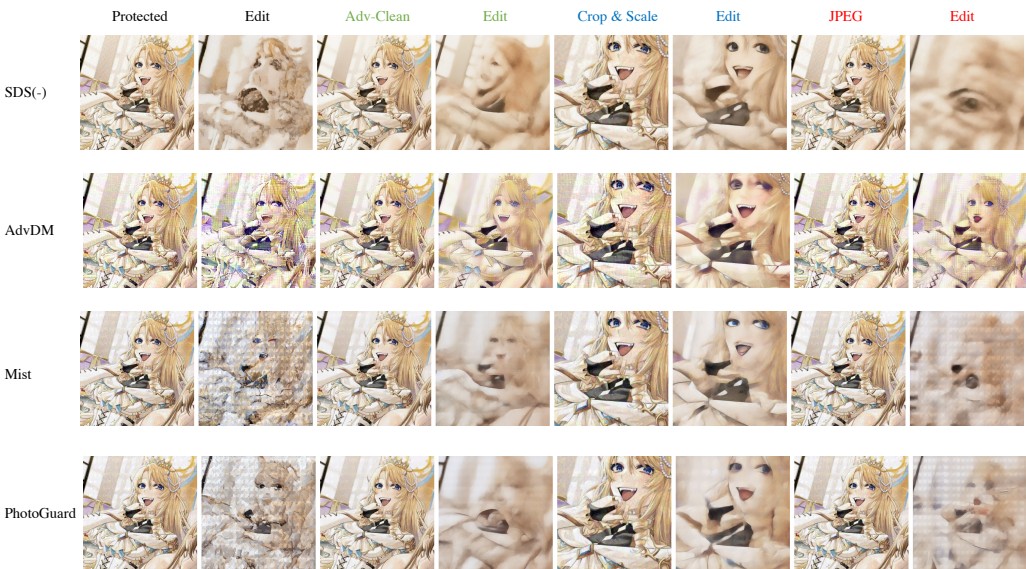

Figure 15: **Protections Against Different Defending Methods**: different rows represent different protection methods, including SDS(-), AdvDM, Mist and PhotoGuard in our proposed design space; each columns show the editing results under different defending methods, including no defending (black), Adv-Clean (green), Crop-and-Resize (blue) and JPEG compression (red).

- JPEG compression: (Sandoval-Segura et al., 2023) reveals that JPEG compression can be a good purification method, and we adopt the $65\%$ as the quality of compression in (Sandoval-Segura et al., 2023).

We show the results of all three defenses in Figure 15 and Figure 16. From the figures, we can find that all three methods fail to fully defend the protection: the cleaned samples can still make the output bad. Among these Crop & Resize seems to be a relatively good defending method. We can also see that AdvDM can be largely purified by Adv-Clean since it contains more high-frequency perturbations. And we also find that SDS(-) is quite robust to all the defending.

## E  BLACKBOX TRANSFERABILITY

Here we show that the protection can be transferred to other popular latent diffusion models. Specifically, we pick some famous publicly-available LDM backbones: SD-V1.4, SD-V1.5 and SD-V2.1. We generate our attacks SD-V1.4 without knowing the parameters of the other two models, playing as blackbox settings. (Zhang et al., 2023c) also shows similar findings when attacking the LDMs.

## F  LOSS CURVE OF SDS VS NO-SDS

In Figure 18 we compare the loss curve of attacks using SDS vs no-SDS. Through the figure, we can see that applying SDS will not significantly change the loss curve, which proves that the Jacobian of the U-Net can be approximated. Applying SDS in attacking an LDM turns out to be a free lunch.

## G  FURTHER EXPLANATION OF SDS LOSS

The SDS loss defined in our settings is:

$$\nabla_x \mathcal{L}_{SDS}(x) = \mathbb{E}_{t,\epsilon}\left[\lambda(t)(\epsilon_\theta(z_t, t) - \epsilon)\frac{\partial z_t}{\partial x}\right] \tag{6}$$

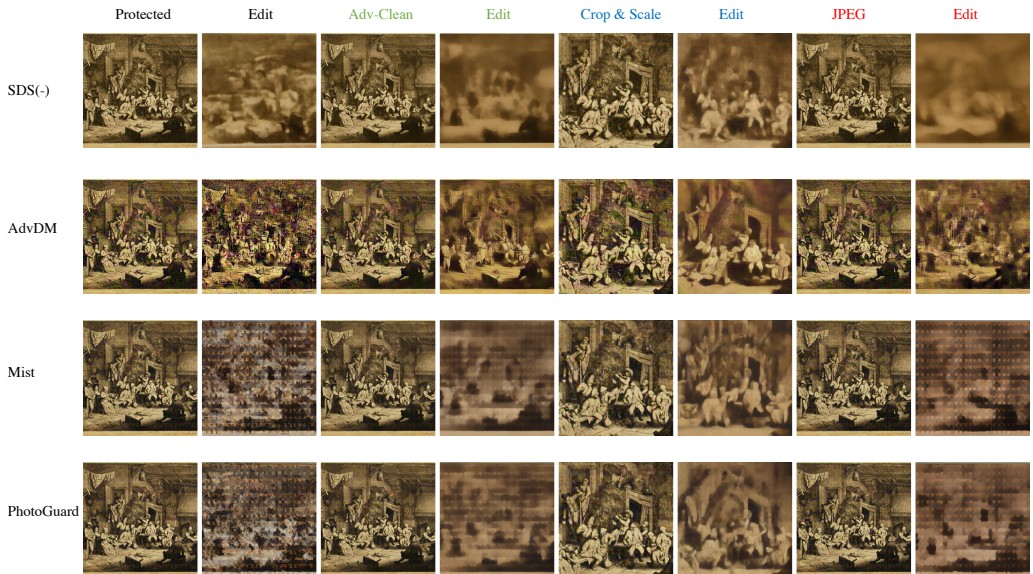

Figure 16: **Protections Against Different Defending Methods**: different rows represent different protection methods, including SDS(-), AdvDM, Mist and PhotoGuard in our proposed design space; each column shows the editing results under different defending methods, including no defending (black), Adv-Clean (green), Crop-and-Resize (blue) and JPEG compression (red).

Intuitively, it can be regarded as approximating the gradient of $\epsilon_\theta(z_t, t)$ over $z_t$ in the Jacobian of U-Net by an identity matrix (Poole et al., 2022).

Meanwhile, it can also be regarded as the weighted probability density distillation loss (Poole et al., 2022):

$$\nabla_x \mathcal{L}_{SDS}(x) = \nabla_x \mathbb{E}_{t,\epsilon}[\mu(t)\lambda(t)\text{KL}(q(z_t|x)\|p_\theta(z_t))] \tag{7}$$

The proof is quite straightforward:

$$\text{KL}(q(z_t|x)\|p_\theta(z_t)) = \mathbb{E}_{t,\epsilon}[\log q(z_t|x) - \log p_\theta(z_t)] \tag{8}$$

$$\nabla_x \text{KL}(q(z_t|x)\|p_\theta(z_t)) = \mathbb{E}_\epsilon[\underbrace{\nabla_x \log q(z_t|x)}_{(A)} - \underbrace{\nabla_x \log p_\theta(z_t)}_{(B)}] \tag{9}$$

where (A) is the gradient of the entropy of the forward process, since the variance is fixed, this entropy is a constant and we have $\nabla_x \log q(z_t|x) = 0$. For the second term (B), we have: $\nabla_x \log p_\theta(z_t) = \nabla_{z_t} \log p_\theta(z_t)\frac{\partial z_t}{\partial x} \approx s_\theta(z_t)\frac{\partial z_t}{\partial x}$ where $s_\theta$ is score function parametrized with $\theta$, which can be transferred to the noise prediction which leads to (B)$= -\mu(t)\epsilon_\theta(z_t, t)\frac{\partial z_t}{\partial x}$.

Finally, since the variable $\epsilon$ has zero-mean, we can use it to reduce the variance (Poole et al., 2022), then we have:

$$\nabla_x \mathcal{L}_{SDS}(x) = \nabla_x \mathbb{E}_{t,\epsilon}\left[\lambda(t)(\epsilon_\theta(z_t, t) - \epsilon)\frac{\partial z_t}{\partial x}\right] = \nabla_x \mathbb{E}_{t,\epsilon}[\mu(t)\lambda(t)\text{KL}(q(z_t|x)\|p_\theta(z_t))] \tag{10}$$

## H DISCUSSION OF WHY MINIMIZING SEMANTIC LOSS CAN WORK

It is an interesting phenomenon that minimizing the semantic loss $\mathcal{L}_S(x)$ counter-intuitively fools the diffusion model, here we provide some insights into the possible reason behind it.

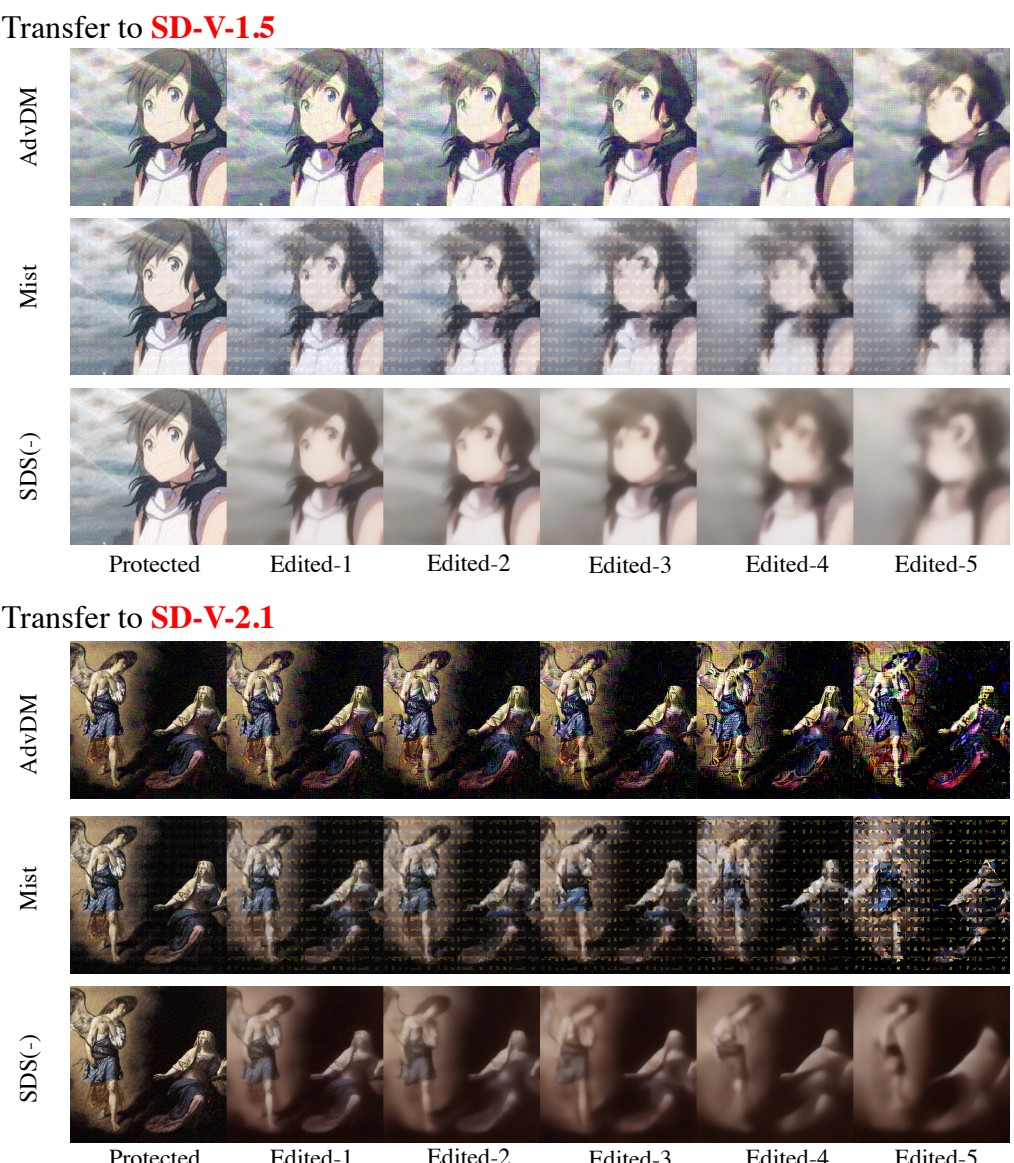

Figure 17: **Blackbox Transferability**: we find that the attacks on SD-V1.4 can be perfectly transferred to other diffusion models such as SD-V1.5 and SD-V2.1.

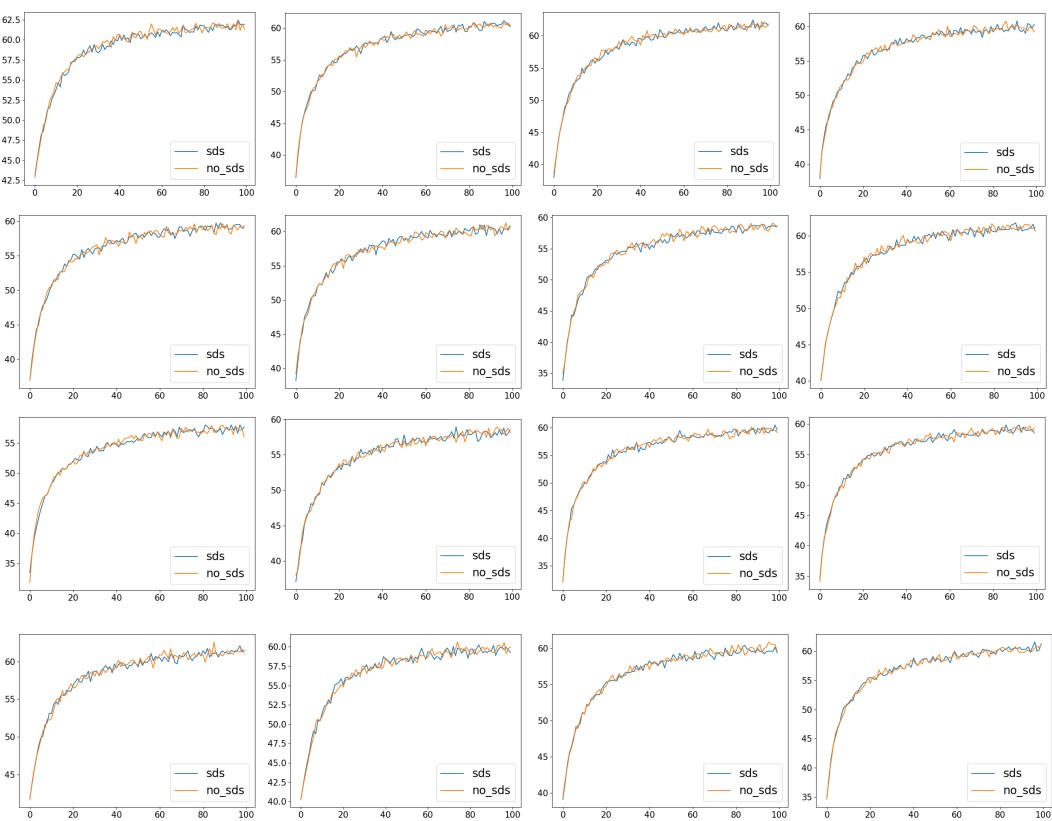

Figure 18: **Loss Curve of SDS vs No-SDS**: we show that applying SDS basically does not change the loss curve, showing that the approximation is practically reasonable. Here we test the loss on fixed timesteps for more stable visualization.

Remember we have the semantic loss in SDS form can be regarded as the weighted probability density distillation loss:

$$\nabla_x \mathcal{L}_{SDS}(x) = \nabla_x \mathbb{E}_{t,\epsilon}[\mu(t)\lambda(t)\text{KL}(q(z_t|x)\|p_\theta(z_t))] \tag{11}$$

Here we can get some insights from the above equation, minimizing the semantic loss means minimizing the KL divergence between $q(z_t|x)$ and $p_\theta(z_t)$, where $p_\theta(z_t)$ is the marginal distribution sharing the same score function learned with parameter $\theta$. Since $p_\theta(z_t)$ does not depend on $x$, it actually describes the learned distribution of the dataset smoothed with Gaussian.

Let's assume in a simple case the data distribution is exactly Dirac Delta Distribution of data points in the dataset. Then we have $p_\theta(z_t)$ as a Gaussian-smoothed composition of Dirac Delta Distribution if $s_\theta$ is perfectly learned. Then minimizing the semantic loss $\mathcal{L}_S(x)$ turns into making $q(z_t|x)$ closer to the Gaussian-smoothed composition of Dirac Delta Distribution, and the optimization direction is to make $x$ closer to the average of data points in the dataset, which brings blurred $x$, which turns out to be a good protection.

## I   HUMAN EVALUATIONS

To better evaluate the quality of perturbation of each protection method, and the strength of protection from a human level, we conducted a survey among humans with the assistance of the Google Form. We got responses from 53 individuals, 70% of them completed the survey on the computer and the rest of them completed the form on their mobile phones.

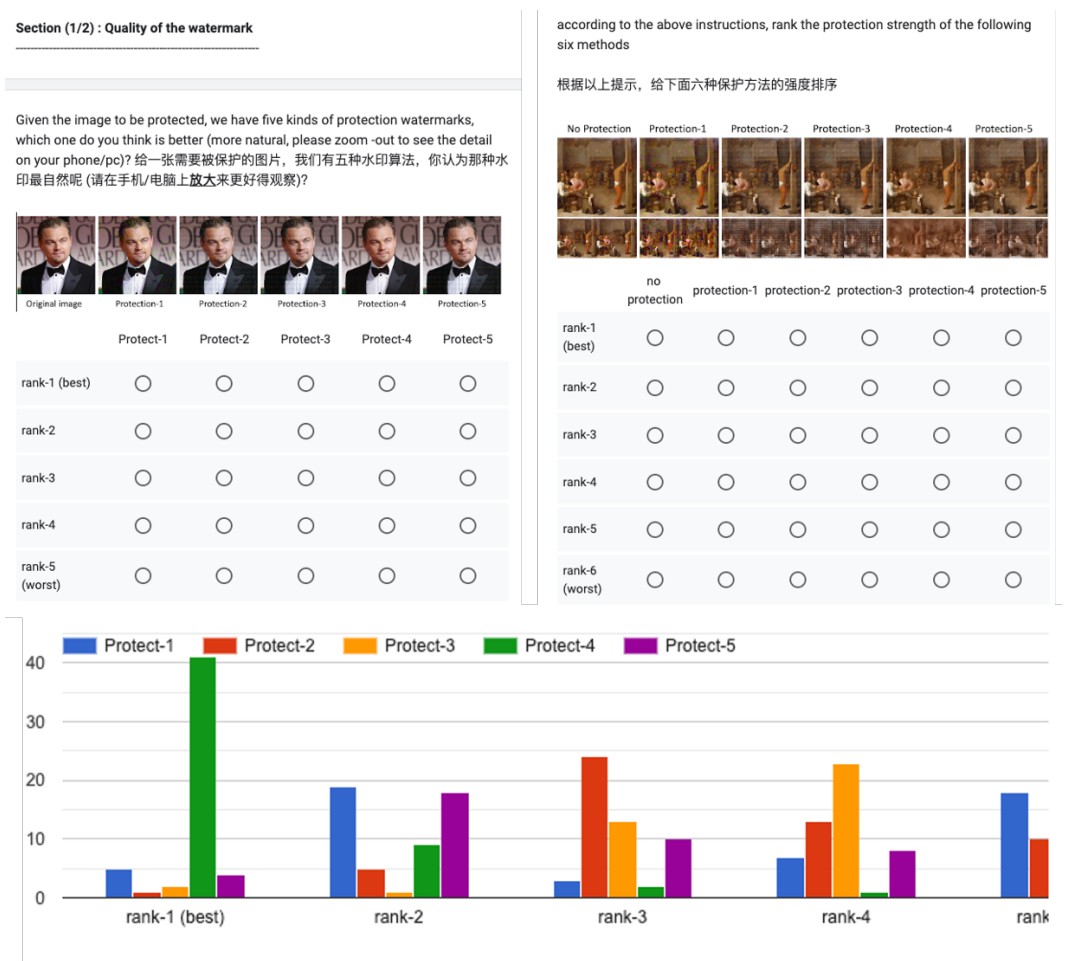

Figure 19: **Survey for Human Evaluation:** We show questions from our survey, the left one is used to evaluate which protection method looks more natural, and the right one is used to evaluate the strength of different protections. The second row demonstrates one statistical result of one question shown in our backstage.

The user interface of the survey is shown in Figure 19, where we have two sections. The first section is used to evaluate the quality of perturbation, and the second section is for finding out the strength of protection from a human's perspective. The participants are asked to rank the given methods. We have 16 questions in total.

The scores are calculated using the rank of each method. For each question, the rank-1 will get 5 points and the lowest rank will get 1 points. The final score in Table 1 and Table 2 are calculated using the average score over all samples.

