# OpenReview forum: "Toward effective protection against diffusion-based mimicry through score distillation"
_ICLR.cc/2024/Conference — ICLR 2024 poster_

### Official Review · Reviewer_7HTv · 2023-10-28

**Soundness:** 2 fair
**Presentation:** 2 fair
**Contribution:** 3 good
**Rating:** 6
**Confidence:** 4

**Summary:**

This paper investigates methods to protect images from unauthorized editing by diffusion models, referred to as "diffusion mimicry". The authors make the insight that the encoder module is more vulnerable than the denoiser when attacking latent diffusion models (LDMs). Based on this finding, they propose faster protection generation using score distillation sampling. They also explore minimizing the semantic loss for protection, obtaining more imperceptible perturbations. The experimental results on divers image datasets support the proposed techniques.

**Strengths:**

This paper reveals that the encoder is more vulnerable than the denoiser, which is novel and insightful.
The proposed SDS is a more efficient protection compared with prior works.

**Weaknesses:**

1. The major concern I have is that how to evaluate the diffusion model still works well after the perturbation? This paper only provides results that the proposed SDS can defend against adversarial attack, but does not show the perturbation does not affect the original function of the diffusion model. The paper lacks both analysis and experiments on this point.

2. The threat model is not clear. Based on my understanding, the core of this paper is adversarial attack and defense on diffusion model. However, the introduction of "diffusion mimicry" confuses the reader a lot. If authors really want to mention "diffusion mimicry", it should be explained that how diffusion model is used for mimicry and why the proposed method is a good defense?

3. The paper lacks insight into why minimizing semantic loss works better.

**Questions:**

Please see the weakness.

---

> ### Author Response · Authors · 2023-11-16
> **Thanks for your valuable comments!**
>
> Thanks for your time to review our paper, thanks for acknowledging our contributions, here let us answer your questions, hopefully it can address your concern.
>
> > The major concern I have is that how to evaluate the diffusion model still works well after the perturbation? This paper only provides results that the proposed SDS can defend against adversarial attack, but does not show the perturbation does not affect the original function of the diffusion model. The paper lacks both analysis and experiments on this point.
>
> I think there may be some misunderstanding about the task, we only do perturbation to the image, and do not change the diffusion model, so it will not `affect original function of the diffusion model`.  SDS is only an alternative loss function, and it can be easily applied without changing the original diffusion model. We are looking forward to get your response if there are still some concerns.
>
> We also clarify our problem in the General Response.
>
>
>
> > The threat model is not clear. Based on my understanding, the core of this paper is adversarial attack and defense on diffusion model. However, the introduction of "diffusion mimicry" confuses the reader a lot. If authors really want to mention "diffusion mimicry", it should be explained that how diffusion model is used for mimicry and why the proposed method is a good defense?
>
>
> Diffusion mimicry is a term used in [1], and here we mention mimicry as malicious editing of a individual image, which is a real-world problem, we ground it to protection against diffusion-based editing in our paper. Our experiments includes three scenarios: global image to image, inpainting and textual-inversion, all can be found in some real-world applications such as face swapping, subject-driven image generative, meme generation, style copying…
>
> In conclusion, just like other works [1, 2, 3], the problem settings are the same, we ground the real-world problem into reasonable settings by attacking the diffusion model.
>
> We also clarify our problem settings again in the General Response.
>
>
>
> > The paper lacks insight into why minimizing semantic loss works better.
>
>
>
> It is an interesting phenomenon that minimizing the semantic loss $\mathcal{L}_{S}(x)$ counter-intuitively fools the diffusion model, here we provide some insights into the possible reasons behind it.
>
> Remember we have the semantic loss in SDS form can be regarded as the weighted probability density distillation loss (Eq.10 in Appendix G).
>
> Here we can get some insights from the equation, minimizing the semantic loss means minimizing the KL divergence between $q(z_t|x)$ and $p_{\theta}(z_t)$, where $p_{\theta}(z_t)$ is the marginal distribution sharing the same score function learned with parameter $\theta$. Since $p_{\theta}(z_t)$ does not depend on $x$, it actually describes the learned distribution of the dataset smoothed with Gaussian.
>
> Let's assume in a simple case that the data distribution is exactly the Dirac Delta Distribution of data points in the dataset. Then we have $p_{\theta}(z_t)$ as a Gaussian-smoothed composition of Dirac Delta Distribution if $s_{\theta}$ is perfectly learned. Then minimizing the semantic loss $\mathcal{L}_{S}(x)$ turns into making $q(z_t|x)$ closer to the Gaussian-smoothed composition of Dirac Delta Distribution, and the optimization direction is to make $x$ closer to the spatially average value in the dataset, which brings blurred $x$, and it turns out to be a good protection.
>
>
>
>
>
> [1] Glaze: Protecting Artists from Style Mimicry by Text-to-Image Models
>
> [2]  Raising the cost of malicious ai-powered image editing.
>
> [3]  Mist: Towards improved adversarial examples for diffusion models.

---

### Official Review · Reviewer_9rgQ · 2023-10-30

**Soundness:** 2 fair
**Presentation:** 3 good
**Contribution:** 3 good
**Rating:** 5
**Confidence:** 3

**Summary:**

The paper aims to generate adversarial examples to protect images from diffusion-based mimicry pipelines, such as image editing, inpainting, and textual inversion. Specifically, the authors explore the robustness of encoder and denoiser modules in the Latent Diffusion Model. And they conclude that the encoder is much more vulnerable than the denoiser. Therefore, they leverage the existing method, Score Distillation Sampling, to calculate the gradient towards minimizing the semantic loss.

**Strengths:**

1. This paper focuses on protecting images from diffusion-based mimicry pipelines, this research region is interesting.
2. Motivated by the observation that the encoder is more vulnerable, the authors resort to Score Distillation Sampling to simplify the gradient update.
3. The authors attack multiple tasks: image editing, inpainting, and textual inversion.

**Weaknesses:**

The major concern is the limited contribution of this paper.
1. The observation that the encoder is more vulnerable than the denoiser only contributes to this region, it is limited in developing more secure LDM. Besides, the experiments empirically indicate this conclusion. It lacks theoretical proofs.
2. The authors resort to the existing method, Score Distillation Sampling,  to fasten the backpropagation of the semantic loss.
3. The authors find that applying gradient descent over the semantic loss leads to more harmonious adversarial images with the original ones. The authors should delve into this phenomenon.

**Questions:**

Given the fact that the encoder is more vulnerable than the denoiser, what is the performance when reducing the gradient of the denoising module?

---

> ### Author Response · Authors · 2023-11-16
> **Thanks for your valuable comments!**
>
> Thanks for your time to review our paper, thanks for acknowledging our contributions, here let us answer your questions, hopefully it can address your concern.
>
>
> > The observation that the encoder is more vulnerable than the denoiser only contributes to this region, it is limited in developing more secure LDM. Besides, the experiments empirically indicate this conclusion. It lacks theoretical proofs.
>
> We believe that `the encoder is more vulnerable` is  a meaningful finding and can contribute new insights to the community to show that the bottleneck of attacking a LDM is the encoder.
>
>
> We further show in the **Appendix C** in revision that attacking a pixel-based DM using semantic loss can not work, which proves that attacking a DM is very different from attacking a LDM, where the sucess of attacking a LDM really comes from the encoder.
>
> We showed a lot of evidences in Section 4 to support our claim,  we do not try to focus on give a theoretical analysis but rather call emphasis on our insights. We write further clarification about our contribution in the **General Response**.
>
>
>
>
>
> > The authors resort to the existing method, Score Distillation Sampling, to fasten the backpropagation of the semantic loss.
>
>
> SDS is a free lunch, but it is fully omitted by previous methods [1,2,3], which turns out to be too expensive to run. We are the first to apply SDS into the protection framework and did extensive experiments to show that it is effective. Also, we provide theoratical analysis to better explain the SDS loss from the perspective of gradient of KL divergence in revision of **Appendix F** in the protection framework.
>
>
> > The authors find that applying gradient descent over the semantic loss leads to more harmonious adversarial images with the original ones. The authors should delve into this phenomenon.
>
> It is an interesting phenomenon that minimizing the semantic loss $\mathcal{L}_{S}(x)$ counter-intuitively fools the diffusion model, here we provide some insights into the possible reasons behind it.
>
> Remember we have the semantic loss in SDS form can be regarded as the weighted probability density distillation loss (Eq.10 in Appendix G).
>
> Here we can get some insights from the equation, minimizing the semantic loss means minimizing the KL divergence between $q(z_t|x)$ and $p_{\theta}(z_t)$, where $p_{\theta}(z_t)$ is the marginal distribution sharing the same score function learned with parameter $\theta$. Since $p_{\theta}(z_t)$ does not depend on $x$, it actually describes the learned distribution of the dataset smoothed with Gaussian.
>
> Let's assume in a simple case that the data distribution is exactly the Dirac Delta Distribution of data points in the dataset. Then we have $p_{\theta}(z_t)$ as a Gaussian-smoothed composition of Dirac Delta Distribution if $s_{\theta}$ is perfectly learned. Then minimizing the semantic loss $\mathcal{L}_{S}(x)$ turns into making $q(z_t|x)$ closer to the Gaussian-smoothed composition of Dirac Delta Distribution, and the optimization direction is to make $x$ closer to the spatially average value in the dataset, which brings blurred $x$, and it turns out to be a good protection.

---

> > ### Author Response · Authors · 2023-11-20
> > **Reviewer-author discussion period ends in Two Days**
> >
> > Thank you again for your review. We hope that we were able to address your concerns in our response. If possible, please let us know if you have any further questions before the reviewer-author discussion period ends. We are glad to address your further concerns, thanks.

---

> > ### Comment · Reviewer_9rgQ · 2023-11-21
> > **Thanks for your responses**
> >
> > I have read your response and other comments carefully.
> > Your responses partly solve my concerns. I have raised my score to 5.
> > However, the semantic loss is the main contribution of this paper. I suggest the authors should provide more theoretical analysis, and include them in the main paper.

---

> > > ### Author Response · Authors · 2023-11-21
> > > **Thanks a lot for your responses**
> > >
> > > Thanks again for your valuable comments and for your patience. We will include the  theoretical analysis we provided during the rebuttal in the main paper, thanks for your suggestions.
> > >
> > > We list our contributions in the **General Response**, actually the semantic loss itself is old, but how to understand it is very critical, the previous works only use it intuitively, and we point out that the encoder actually matters, we even further show that using semantic loss only on pixel-based diffusion models cannot even work (in the revision of **Appendix C**)!  This insight is important for the community to better understand this task, where latent space in the LDM is the key.

---

### Official Review · Reviewer_vkfi · 2023-10-30

**Soundness:** 3 good
**Presentation:** 2 fair
**Contribution:** 3 good
**Rating:** 6
**Confidence:** 4

**Summary:**

This paper wants to protect images from unauthorized use by perturbing the images using projected gradient descent. It uses the existing score distillation sampling idea to approximate the gradient of the sample instead of backward propagation through the UNet to save resources and time. It also chooses to minimize the semantic loss (ie, the diffusion model's loss) instead of maximizing it to generate more natural perturbations. Experiments on different domains and tasks show it outperforms the existing baselines.

**Strengths:**

1. This paper targets an important issue of preventing unauthorized image use.
2. Reducing the time cost and resource requirement makes the method more affordable and practical.
3. Experiments show it's better than existing baselines.

**Weaknesses:**

1. From my understanding, this method needs the gradient from the misused models. That is, the protection perturbation is generated on the exact model used to maliciously manipulate the image. It's not clear if this generated perturbation is only effective on that model, or if it can protect the image from misuse by an unknown model.

2. No defense methods are evaluated. The malicious people may apply image transformations to remove the adversarial perturbations.

3. It's not clear why minimizing the loss (encouraging LDM to make better predictions) can protect the images. It's very counter-intuitive and needs a better explanation. I can understand that minimizing the loss with respect to certain targets as Photoguard can work because it encourages the LDM to generate an image different from the original image to protect. But why can encouraging the LDM to generate a better image of the original image (the opposite goal of AdvDM) can help? Maybe it's kind of trapping the sample into a local optimum so that the later optimization/editing is also stuck? But this also needs to be justified.

4. In Section 4.3, the conclusion of denoiser being more robust needs more evidence. Figure 2 shows the $\delta_z$ can be less than 10x $\delta_x$ for many cases. So I think the 10x budget (Figure 10 has even larger budgets) should be large enough to conduct some successful attacks. It's not clear why they fail.

5. Photoguard has a diffusion attack. Why is only the encoder attack mentioned in the related work and evaluated in the experiments?

6. The detailed setup for the experiments is missing. For example, how many iterations are used for the baselines and proposed method? What is the learning rate? What's the perturbation budget? How large is the fixed $\delta_x$ in Figure 2?

7. The pseudocode in Appendix A doesn't allow `x` to have gradients because `x=x.detach().clone()` detaches `x` from the computational graph.

8. I suggest briefly introducing the metrics and human studies and having a more detailed explanation of the experimental results in the main text.

9. To show the effectiveness of SDS approximation, one may visualize the loss changes and the gradient landscapes with the original loss function and the SDS loss.

**Questions:**

1. Can this protection effect transfer to different LDMs? That is, assume the image is protected using a LDM A, when the adversary uses another LDM B to edit the image, will it fail?
2. Can image transformations such as JPEG compression, cropping, rotation, and rescaling invalidate the protection?
3. Why can minimizing the loss (encouraging LDM to make better predictions) protect the images?
4. How are the $z$ and $z_{adv}$ visualized in the figures such as Figure 2?
5. Could you explain, why attacking the diffusion model via $\mathcal{L}_S$ outperforms $\mathcal{L}_T$ if the encoder is less robust than the diffusion model as claimed in the paper?

**Details Of Ethics Concerns:**

This paper has user studies. Not clear if the authors have got the IRB approval.

---

> ### Author Response · Authors · 2023-11-16
> **Thank you so much for thoroughly reviewing our paper!  [Part (1)]**
>
> Thank you so much for thoroughly reviewing our paper, your careful attention to detail is greatly appreciated. We are happy to answer some of your questions and we hope that we can address your concerns:
>
>
> > From my understanding, this method needs the gradient from the misused models. That is, the protection perturbation is generated on the exact model used to maliciously manipulate the image. It's not clear if this generated perturbation is only effective on that model, or if it can protect the image from misuse by an unknown model.
>
> That is a good question. Here we focus on the **white-box settings** the same as previous works  [1, 2, 3]. For the black-box settings as the reviewer mentioned, recent paper [4] found that the adversarial samples can transfer between the most popular publicly-avaible LDMs like SD-v1.4, SD-v1.5 and SD-v2. We also did experiments to show that the adversarial samples generated on the basic SD-v1.4 can be transferred to SD-v1.5 and SD-v2. We put the results in Appendix E of the revision following your suggestion.
>
>
> > No defense methods are evaluated. The malicious people may apply image transformations to remove the adversarial perturbations.
>
> We test some famous defending methods, including JPEG compression, AdverseClean (https://github.com/lllyasviel/AdverseCleaner) and crop-and-rescale on the adversarial samples. We put additional results in Appendix D following your suggestion, we found that crop-and-resize turns out to be relatively effective, but it still cannot work well to remove the perturbations. This finding is in accordance with that also found in [3].
>
>
>
> > It's not clear why minimizing the loss (encouraging LDM to make better predictions) can protect the images. It's very counter-intuitive and needs a better explanation. I can understand that minimizing the loss with respect to certain targets as Photoguard can work because it encourages the LDM to generate an image different from the original image to protect. But why can encouraging the LDM to generate a better image of the original image (the opposite goal of AdvDM) can help? Maybe it's kind of trapping the sample into a local optimum so that the later optimization/editing is also stuck? But this also needs to be justified.
>
>
>
>
> It is an interesting phenomenon that minimizing the semantic loss $\mathcal{L}_{S}(x)$ counter-intuitively fools the diffusion model, here we provide some insights into the possible reasons behind it.
>
> Remember we have the semantic loss in SDS form can be regarded as the weighted probability density distillation loss (Eq.10 in Appendix G).
>
> Here we can get some insights from the equation, minimizing the semantic loss means minimizing the KL divergence between $q(z_t|x)$ and $p_{\theta}(z_t)$, where $p_{\theta}(z_t)$ is the marginal distribution sharing the same score function learned with parameter $\theta$. Since $p_{\theta}(z_t)$ does not depend on $x$, it actually describes the learned distribution of the dataset smoothed with Gaussian.
>
> Let's assume in a simple case that the data distribution is exactly the Dirac Delta Distribution of data points in the dataset. Then we have $p_{\theta}(z_t)$ as a Gaussian-smoothed composition of Dirac Delta Distribution if $s_{\theta}$ is perfectly learned. Then minimizing the semantic loss $\mathcal{L}_{S}(x)$ turns into making $q(z_t|x)$ closer to the Gaussian-smoothed composition of Dirac Delta Distribution, and the optimization direction is to make $x$ closer to the spatially average value in the dataset, which brings blurred $x$, and it turns out to be a good protection.
>
>
>
>
>
>
> > In Section 4.3, the conclusion of denoiser being more robust needs more evidence. Figure 2 shows that $\delta_z$  can be less than 10x$\delta_x$ for many cases. So I think the 10x budget (Figure 10 has even larger budgets) should be large enough to conduct some successful attacks. It's not clear why they fail.
>
>
> That is a good question. This exactly supports our claim that the bottleneck is the encoder, when attacking the z-space alone, we do not use the gradient from the encoder, which brings a weak attack to the encoder itself. As we can see in Figure 10, although the budgets are getting even larger, the edited results do not diverge a lot from x_adv (which may be quite noisy since we perturb the z-space).
>
>
> In conclusion, even given the same budget, simply attacking the z-space can not give us a good direction to attack the LDM. Encoder is really important and is the key when attacking a LDM.
>
>
> To better support our conclusion, we did experiments (in **Appendix C** in revision) on a pixel-based diffusion model without encoder-decoder, and we found that gradient-based methods fail to work. That is, if we attack the DM using PGD and semantic loss, the generated adversarial sample cannot even fool the DM, the reason behind this may be the randomness brought by Gaussian noise term, making the denoiser quite robust to be attacked directly.

---

> > ### Author Response · Authors · 2023-11-16
> > **Thank you so much for thoroughly reviewing our paper! [Part (2)]**
> >
> > > Photoguard has a diffusion attack. Why is only the encoder attack mentioned in the related work and evaluated in the experiments?
> >
> >
> > The diffusion attack in Photoguard is an end-to-end attack against SDEdit-based inpainting, which means we need to final edit results over the inputs, which contain the chain of U-Net. It will take like 50G and even more on our settings, which is actually not a practical attack, also in previous works [1, 3] we only compare the encoder attack in PhotoGuard.
> >
> >
> >
> >
> > > The detailed setup for the experiments is missing. For example, how many iterations are used for the baselines and proposed method? What is the learning rate? What's the perturbation budget? How large is the fixed $\delta_x$ in Figure 2?
> >
> >
> > For all the methods (baselines and our method) we use the PGD attack to generate adversarial samples, and we use $100$ as the number of timesteps and $\delta=16/255$ (fixed $\delta$ in Figure 2.) and step-size as $1/255$. We put this in the revision in **Appendix B.2**.
> >
> >
> > > The pseudocode in Appendix A doesn't allow x to have gradients because x=x.detach().clone() detaches x from the computational graph.
> >
> >
> > Thanks for your careful reading. We miss `x.requires_grad=True` to enable the gradient calculation in the pseudo-code. We have fixed this in the revision.
> >
> >
> >
> >
> > > I suggest briefly introducing the metrics and human studies and having a more detailed explanation of the experimental results in the main text.
> >
> >
> > We are glad to consider this suggestion in our final version and add more explanation of the experimental in the main text and move some quantitative results to the appendix for enough space.
> >
> >
> >
> >
> > > To show the effectiveness of SDS approximation, one may visualize the loss changes and the gradient landscapes with the original loss function and the SDS loss.
> >
> >
> > That is a good suggestion, we followed it and visualized the loss function in **Appendix F** of the new revision.
> > Also, we provide theoretical analysis to better explain the SDS loss from the perspective of the gradient of KL divergence in the revision of **Appendix F**. We borrowed the idea from [5] to write the proof.
> >
> >
> > > How are $z$ and $z_{adv}$ visualized in figures such as Figure 2?
> >
> > The size of the z-space in a diffusion model is actually 4*64*64, we visualize the first three channels in RGB in figures such as Figure 2, which can already effectively show the large differences in the z-space.
> >
> >
> > > Could you explain, why attacking the diffusion model via $L_{s}$ outperforms $L_{T}$ if the encoder is less robust than the diffusion model as claimed in the paper?
> >
> >
> > Because attacking the diffusion model via $L_{s}$  relies on both the encoder and the denoiser, where the gradient is accumulated on these two modules, which can tell the encode how to generate adv-samples in the z-space to fool the denoiser. While  $L_{T}$ aims to drag the sample in the z-space closer to the target sample, while attacking the encoder can also work well, it knows nothing about the denoiser, which cannot accumulate the error through the iterative noise-then-denoise in the editing framework.
> >
> >
> >
> >
> >
> >
> >
> >
> >
> >
> >
> >
> >
> >
> >
> >
> >
> >
> >
> >
> >
> >
> >
> >
> >
> >
> >
> >
> >
> >
> >
> > [1]  Adversarial example does good: Preventing painting imitation from
> > diffusion models via adversarial examples.
> >
> > [2]  Raising the cost of malicious ai-powered image editing.
> >
> > [3]  Mist: Towards improved adversarial examples for diffusion models.
> >
> > [4] On the Robustness of Latent Diffusion Model
> >
> >
> > [5] DreamFusion: Text-to-3D using 2D Diffusion

---

> ### Author Response · Authors · 2023-11-20
> **Reviewer-author discussion period ends in Two Days**
>
> Thank you again for your really careful review. We hope that we were able to address your concerns in our response. If possible, please let us know if you have any further questions before the reviewer-author discussion period ends. We are glad to address your further concerns, thanks.

---

> > ### Comment · Reviewer_vkfi · 2023-11-21
> > **Thank you for your detailed response.**
> >
> > Thank you very much for your detailed response. I am satisfied with the answers and new illustrations, except for one thing about the photoguard.
> >
> > I ran [the photoguard diffusion attack](https://github.com/MadryLab/photoguard/blob/main/notebooks/demo_complex_attack_inpainting.ipynb) locally and it only took 27G GPU memory. It's much less than the claimed 50G. Could you verify your claim?

---

> ### Comment · Reviewer_vkfi · 2023-11-21
> **I have raised my score, but photoguard needs to be correctly discussed.**
>
> I have raised my score.
>
> But I suggest correcting the discussion about photoguard in the paper (because they do have an encoder attack and a diffusion attack) and talking about why only the simple encoder attack is evaluated, especially double-checking the claim regarding the +50G memory request.

---

> ### Author Response · Authors · 2023-11-21
> **Thanks again for your really careful review and meaningful discussions.**
>
> We are glad that we have solved most of your concerns and thank you for raising the score. We have added a clarification about Photoguard in our main paper in the new revision of **Section 2**.
>
> For the memory consumption of Photoguard, in the early stage of this project, we found that when we use `num_inference_steps=8` in the demo provided by Photoguard, it will take up `~47G`  on one A6000 GPU.
>
> I just tried if we set `num_inference_steps=4` as default, it will take up `29G` on one A6000 GPU, and 30 minutes for one image, which is also too expensive. In conclusion, the diffusion attack of the Photoguard is too expensive to run. We also tried to run it on one 3090 and it did not work.
>
> Also, in the official repo of photoguard(https://github.com/MadryLab/photoguard/blob/main/notebooks/demo_complex_attack_inpainting.ipynb), it said: `Below is the implementation of an end2end attack on the stable diffusion pipeline using PGD. This requires a GPU Memory >= 40Gbs (we ran on 1 A100)`.
>
> Thanks again for your really careful review and meaningful discussions.

---

### Official Review · Reviewer_hmR8 · 2023-11-07

**Soundness:** 2 fair
**Presentation:** 3 good
**Contribution:** 3 good
**Rating:** 6
**Confidence:** 4

**Summary:**

The paper introduced a faster attack method for encoder-diffusion style model. They find out that the encoder is a weak point of the whole model. Therefore, they specifically design an attack method by removing the gradient propagation from the diffusion parts. It makes the running speed faster. They evaluate against three latest methods and show competitive results.

**Strengths:**

+ There are some interesting observations including the different robustness regarding diffusion and encoding parts.
I believe this is a good observation which could make attack fast.
+ The methods are effective against latest benchmarks.

**Weaknesses:**

- Missing details when comparing the magnitude of embedding change versus input change
The authors use absolute value to compare adversarial noise. While the embedding can be of different scale. The authors shall provide a relative scale of perturbation magnitude.

- Unclear math derivation
In equation 5, why can we approximately remove the gradient parts from denoiser?

**Questions:**

The reason behind equation 5 and the relative scale of perturbation.

---

> ### Author Response · Authors · 2023-11-16
> **Thanks for your valuable comments!**
>
> We express our gratitude to the reviewer for conducting a careful review and providing valuable feedback. We highly appreciate the reviewer's recognition of the insights we provided and the effectiveness of our method.
>
> > Missing details when comparing the magnitude of embedding change versus input change The authors use absolute value to compare adversarial noise. While the embedding can be of different scale. The authors shall provide a relative scale of perturbation magnitude.
>
> As we mentioned in Sec 4.1: line 5, we normalized the perturbations in the two spaces for comparison. Specifically, we scale the perturbations in the z-space back to [0, 1] as in the x-space.
>
> > Unclear math derivation In equation 5, why can we approximately remove the gradient parts from the denoiser?
>
>
> This is a good question. Approximating the jacobian of U-Net as an identity matrix is used in many works, such as distillation from 2D [1, 2] and gradient approximation in adversarial attack [3]. While approximation is the most straightforward explanation, it can also be regarded as the weighted probability density distillation loss, we provide detailed analysis in **Appendix G** in the revision.
>
> Also, by visualizing the loss curve in **Appendix F**, we better support the claim and it turns out that using SDS loss is a free lunch that should be noticed and applied when designing a protection method.
>
>
> [1] DreamFusion: Text-to-3D using 2D Diffusion
>
> [2] ProlificDreamer: High-Fidelity and Diverse Text-to-3D Generation with Variational Score Distillation
>
> [3] Diffusion-based adversarial sample generation for improved stealthiness and controllability

---

> ### Author Response · Authors · 2023-11-20
> **Reviewer-author discussion period ends in Two Days**
>
> Thank you again for your review. We hope that we were able to address your concerns in our response. If possible, please let us know if you have any further questions before the reviewer-author discussion period ends. We are glad to address your further concerns, thanks.

---

### Author Response · Authors · 2023-11-16
**General Response**

We thank the reviewers for their valuable comments on our paper. We are excited to see that all the reviewers (Reviewer hmR8, vkfi, 9rgQ, 7HTv) identified our novel insights about the vulnerability of the encoder in an LDM, acknowledging the effectiveness of our proposed SDS attack (Reviewer hmR8, vkfi, 9rgQ) against the baseline methods in various tasks.

We focus on answering some commonly asked questions:


> Q1: Why we can approximate the jacobian of U-Net? (Reviewer hmR8, vkfi)

Approximating the jacobian of U-Net as an identity matrix is used in many works, such as distillation from 2D [1, 2] and gradient approximation in adversarial attack [3]. While approximation is the most straight-forward explanation, it can also be regarded as the weighted probability density distillation loss, we provide detailed analysis in **Appendix G** in the revision.

Also, by visualizing the loss curve in **Appendix F**, we better support the claim about and it turns out that using SDS loss is a free lunch that should be noticed and applied when designing a protection method.




> Q2: Why can minimizing the semantic loss work, how to explain it? (Reviewer vkfi, 9rgQ, 7HTv)


It is an interesting phenomenon that minimizing the semantic loss $\mathcal{L}_{S}(x)$ counter-intuitively fools the diffusion model, here we provide some insights into the possible reasons behind it.

Remember we have the semantic loss in SDS form can be regarded as the weighted probability density distillation loss (Eq.10 in Appendix G).

Here we can get some insights from the equation, minimizing the semantic loss means minimizing the KL divergence between $q(z_t|x)$ and $p_{\theta}(z_t)$, where $p_{\theta}(z_t)$ is the marginal distribution sharing the same score function learned with parameter $\theta$. Since $p_{\theta}(z_t)$ does not depend on $x$, it actually describes the learned distribution of the dataset smoothed with Gaussian.

Let's assume in a simple case that the data distribution is exactly the Dirac Delta Distribution of data points in the dataset. Then we have $p_{\theta}(z_t)$ as a Gaussian-smoothed composition of Dirac Delta Distribution if $s_{\theta}$ is perfectly learned. Then minimizing the semantic loss $\mathcal{L}_{S}(x)$ turns into making $q(z_t|x)$ closer to the Gaussian-smoothed composition of Dirac Delta Distribution, and the optimization direction is to make $x$ closer to the spatially average value in the dataset, which brings blurred $x$, and it turns out to be a good protection.




> Q3: Clarification of contribution and problem settings. (Reviewer vkfi, 9rgQ)

First, we want to clarify our problem settings, especially what is the **attack** and **protection** mentioned in our paper (Reviewer vkfi, 9rgQ): we try to **protect** the images from diffusion-based mimcry, and the way to **protect** them is grounded to generate adversarial samples to **attack** the diffusion model. So the only thing we did is adding perturbations to the image to protect it **without modifying** the diffusion model.

For the contributions, we offer some **novel insights of the bottleneck** of the current white-box protection method against LDM-based mimicry (generating adv-samples for LDM), where we found that the the encoder is actually the bottleneck when attacking an LDM. This is a novel finding that is critical to understanding the adv-samples of LDM.

Then we are **the first to use SDS loss in protection**, showing that SDS is a free lunch for protection, which can help the user dramatically save their cost, and it is **omitted** by the previous protection method.

Finally, we surprisingly find that minimizing the semantic loss of an LDM brings quite good protection. This is slightly counter-intuitive but can be explainable, we put some analysis in the revision.

We believe the three points above will bring meaningful insights into the critical area of protecting against diffusion model. Also, it can inspire further works to discuss deeper into the reason behind it, such as why LDM is so vulnerable to being attacked, whether is there any way to get a robust LDM.


[1] DreamFusion: Text-to-3D using 2D Diffusion

[2] Delta Denoising Score

[3] Diffusion-based adversarial sample generation for improved stealthiness and controllability

---

> ### Comment · Reviewer_7HTv · 2023-11-20
> **About the third bullet (Please number your answers): Clarification of contribution and problem settings.**
>
> The authors do not clarify the problem settings.

---

> > ### Author Response · Authors · 2023-11-20
> > **The problem settings**
> >
> > Problem settings were put before the first bullet. We now reorganized it and numbered our answers.

---

> > > ### Comment · Reviewer_7HTv · 2023-11-20
> > > **Thanks for your quick response. But I am still confused about the problem setting and threat model.**
> > >
> > > 1. Why 'the way to protect them is grounded to generate adversarial samples to attack the diffusion model'?
> > >
> > > 2. Based on my understanding, the perturbation is added during the sampling of diffusion model? Are there experiments you could present to show that the original function of diffusion model is not affected after perturbation? I am ok if you just describe the experiments without showing real results for now.
> > >
> > > 3. One sentence confused me a lot is the first sentence in Section 4:
> > > 'The protection against diffusion-based mimicry is one type of attack against the LDMs.' I may better understand this sentence if you could provide a clear answer to my question 1.
> > >
> > > More questions for you to understand my confusion:
> > > 4. Can you provide an example showing what diffusion model is originally used for? e.g., for generating image
> > >
> > > 5. Then, what attacker will do? e.g., leverage the diffusion model to discover training data.
> > >
> > > 6. How to protect the diffusion model? e.g., adding perturbation to the prompt? When you say adding perturbation to the image, what is the image refers to? What about the text-image diffusion models?

---

> > > > ### Author Response · Authors · 2023-11-20
> > > > **Thanks for your response**
> > > >
> > > > I am glad to answer your questions here, hopefully, it can address your concerns:
> > > >
> > > > > Q1: Why 'the way to protect them is grounded to generate adversarial samples to attack the diffusion model'?
> > > >
> > > > e.g. We have a photo to be protected, now the way we do the protection is to perturb the photo to generate an adversarial sample for the LDM, so that if the mimickers(attackers) use the LDM to edit our photo, they will get some bad outputs. And the perturbation itself is actually attacking the LDM to fool it into generating bad outputs, but the perturbation is used to protect our image.
> > > >
> > > > > Q2: Based on my understanding, the perturbation is added during the sampling of diffusion model? Are there experiments you could present to show that the original function of diffusion model is not affected after perturbation? I am ok if you just describe the experiments without showing real results for now.
> > > >
> > > >
> > > > There may be some misunderstanding. The perturbations are **added directly to the image to be protected**, and the pipeline of the LDM remains the same.
> > > >
> > > > > Q3: One sentence confused me a lot is the first sentence in Section 4: 'The protection against diffusion-based mimicry is one type of attack against the LDMs.' I may better understand this sentence if you could provide a clear answer to my question 1.
> > > >
> > > > Please refer to Q1
> > > >
> > > > > Q4: Can you provide an example showing what diffusion model is originally used for? e.g., for generating image
> > > >
> > > > The diffusion model has been widely used to edit images, e.g. using the SDEdit, inpainting, or textual-inversion pipeline, when the edition is malicious and unauthorized, it becomes a mimicry. Generated images can be seen in Figures 6, 7, 8 for `no protection` cases. If you want to check more generated images, you can also turn to other papers like [1, 2, 3] to see more editing results.
> > > >
> > > > > Q5: Then, what attacker will do? e.g., leverage the diffusion model to discover training data.
> > > >
> > > > The attackers will edit the unprotected image without any permission using diffusion model.
> > > >
> > > > > Q6: How to protect the diffusion model? e.g., adding perturbation to the prompt? When you say adding perturbation to the image, what is the image refers to? What about the text-image diffusion models?
> > > >
> > > > We are not going to protect the diffusion model, we are going to protect our images from being maliciously edited by the diffusion models. The images can be our images shared on the internet (e.g. artworks, selfies). Text-image diffusion models can also be well prevented from conducting malicious editing using gradient-based strategies to perturb our images which are well-studied in [1, 4].
> > > >
> > > >
> > > > [1] Adversarial example does good: Preventing painting imitation from diffusion models via adversarial examples.
> > > >
> > > > [2] Raising the cost of malicious ai-powered image editing.
> > > >
> > > > [3] Mist: Towards improved adversarial examples for diffusion models.
> > > >
> > > > [4] On the Robustness of Latent Diffusion Model
> > > >
> > > > [5] DreamFusion: Text-to-3D using 2D Diffusion

---

> > > > > ### Comment · Reviewer_7HTv · 2023-11-20
> > > > >
> > > > > Thank you very much for your patience and detailed clarifications. My concerns have been fully addressed. After considering the questions raised by other reviewers and your thorough rebuttal, I have decided to increase my score.

---

> > > > > > ### Author Response · Authors · 2023-11-20
> > > > > >
> > > > > > We are glad that all the concerns are solved, thank you for your patience too.

---

### Author Response · Authors · 2023-11-16
**Revision Summary**

## Revision Summary

We moved the appendix part from supplementary material to the main pdf, so you can directly open the link to read the main paper and the appendix without downloading and unzipping the file.

The summary of the latest revision is listed below:


### More experiments:
- Experiments about the blackbox settings: transferability of the protections to different diffusion models are added in **Appendix E**
- We add experiments of defending in **Appendix D**
- We visualize the loss curve of SDS vs no-SDS in **Appendix F**
- Attack pixel-based diffusion model instead of LDM in **Appendix C**


### Clarity:
- We fix the error in pseudo-code in **Appendix A**
- We add detailed settings of the experiments in **Appendix B.2**
- We add a detailed theoretical analysis of SDS approximation in **Appendix  G**
- We discuss why minimizing semantic loss can work and offer some insights in  **Appendix H**

---

### Meta-Review · Area_Chair_evFz · 2023-12-06

**Metareview:**

3x BA and 1x BR. This paper proposes to use score distillation sampling to perturb images with projected gradient descent, so as to protect images from unauthorized uses. The reviewers consistently appreciate their (1) important topic, (2) interesting observations, (3) extensive attacks, and (4) SOTA results. The rebuttal has addressed most of their concerns. Some minor concerns like insufficient theoretical analysis are, however, beyond the scope of this work. The AC therefore leans to accept this submission.

**Justification For Why Not Higher Score:**

N/A

**Justification For Why Not Lower Score:**

N/A

---

### Decision · Program_Chairs · 2024-01-16

Accept (poster)